# ReSync: Riemannian Subgradient-based Robust Rotation Synchronization

**Huikang Liu**
School of Information Management and Engineering
Shanghai University of Finance and Economics
liuhuikang@shufe.edu.cn

**Xiao Li**[*]
School of Data Science
The Chinese University of Hong Kong, Shenzhen
lixiao@cuhk.edu.cn

**Anthony Man-Cho So**
Department of Systems Engineering and Engineering Management
The Chinese University of Hong Kong
manchoso@se.cuhk.edu.hk

## Abstract

This work presents ReSync, a Riemannian subgradient-based algorithm for solving the robust rotation synchronization problem, which arises in various engineering applications. ReSync solves a least-unsquared minimization formulation over the rotation group, which is nonsmooth and nonconvex, and aims at recovering the underlying rotations directly. We provide strong theoretical guarantees for ReSync under the random corruption setting. Specifically, we first show that the initialization procedure of ReSync yields a proper initial point that lies in a local region around the ground-truth rotations. We next establish the weak sharpness property of the aforementioned formulation and then utilize this property to derive the local linear convergence of ReSync to the ground-truth rotations. By combining these guarantees, we conclude that ReSync converges linearly to the ground-truth rotations under appropriate conditions. Experiment results demonstrate the effectiveness of ReSync.

## 1 Introduction

Rotation synchronization (RS) is a fundamental problem in many engineering applications. For instance, RS (also known as "rotation averaging") is an important subproblem of structure from motion (SfM) and simultaneous localization and mapping (SLAM) in computer vision [20, 22, 17], where the goal is to compute the absolute orientations of objects from relative rotations between pairs of objects. RS has also been applied to sensor network localization [41, 12], signal recovery from phaseless observations [2], digital communications [36], and cryo-EM imaging [35, 33].

Practical measurements of relative rotations are often *incomplete* and *corrupted*, leading to the problem of robust rotation synchronization (RRS) [28, 21, 22, 39, 9, 31]. The goal of RRS is to reconstruct a set of ground-truth rotations $X_1^\star, \cdots, X_i^\star, \cdots, X_n^\star \in \mathrm{SO}(d)$ from measurements of

---

[*]Corresponding Author

37th Conference on Neural Information Processing Systems (NeurIPS 2023).

relative rotations represented as

$$
\boldsymbol{Y}_{ij} = \begin{cases} \boldsymbol{X}_i^\star \boldsymbol{X}_j^{\star\top}, & (i,j) \in \mathcal{A}, \\ \boldsymbol{O}_{ij}, & (i,j) \in \mathcal{E} \setminus \mathcal{A}, \\ \boldsymbol{0}, & (i,j) \in \mathcal{E}^c, \end{cases} \quad \text{with} \quad (i,j) \in \begin{cases} \mathcal{A}, & \text{with ratio } pq, \\ \mathcal{E} \setminus \mathcal{A}, & \text{with ratio } (1-p)q, \\ \mathcal{E}^c, & \text{otherwise.} \end{cases} \tag{1}
$$

Here, $\mathrm{SO}(d) := \left\{ \boldsymbol{R} \in \mathbb{R}^{d \times d} : \boldsymbol{R}^\top \boldsymbol{R} = \boldsymbol{I}, \det(\boldsymbol{R}) = 1 \right\}$ denotes the rotation group (also known as the special orthogonal group), $\mathcal{E}$ represents the indices of all available observations, $\mathcal{A}$ denotes the indices of true observations, $\mathcal{A}^c := \mathcal{E} \setminus \mathcal{A}$ is the indices of *outliers*, $\boldsymbol{O}_{ij} \in \mathrm{SO}(d)$ is an outlying observation, and the missing observations are set to be $\boldsymbol{0}$ by convention; see, e.g., [23, section 2.1]. We use $q \in (0,1)$ to denote the observation ratio and $p \in (0,1)$ to denote the ratio of true observations.

**Related works.** Due to the vast amount of research in this field, our overview will necessarily focus on theoretical investigations of RS. In the case where no outliers exist in the measurement model (1), i.e., $p = 1$, a natural formulation is to minimize a smooth least-squares function $\sum_{(j,j) \in \mathcal{E}} \| \boldsymbol{X}_i \boldsymbol{X}_j^\top - \boldsymbol{Y}_{ij} \|_F^2$ over $\boldsymbol{X}_i \in \mathrm{SO}(d)$, $1 \le i \le n$. Spectral relaxation and semidefinite relaxation (SDR) are typical approaches for addressing this problem [34, 3, 6, 5, 4, 30], where they provide strong recovery guarantees. However, these results cannot be directly applied to the corrupted model (1) due to the existence of outliers (i.e., $p < 1$) and the sensitivity of the least-squares solution to outlying observations.

Theoretical understanding of RRS is still rather limited. One typical setting for theoretical analysis of RRS is the random corruption model (RCM); see Section 2.2. The work [39] introduces a least-unsquared formulation and applies the SDR method to tackle it. Under the RCM and in the full observation case where $q = 1$, it is shown that the minimizer of the SDR reformulation exactly recovers the underlying Gram matrix (hence the ground-truth rotations) under the conditions that the true observation ratio $p \ge 0.46$ for $\mathrm{SO}(2)$ (and $p \ge 0.49$ for $\mathrm{SO}(3)$) and $n \to \infty$. In [23], the authors established the relationship between cycle-consistency and exact recovery and introduced a message-passing algorithm. Their method is tailored to find the corruption level in the graph, rather than recovering the ground-truth rotations directly. They provided linear convergence guarantees for their algorithm once the ratios satisfy $p^8 q^2 = \Omega(\log n / n)$ under the RCM. However, it is unclear how this message-passing algorithm is related to other optimization procedures for solving the problem. Let us mention that they also provided guarantees for other compact groups and corruption settings. Following partly the framework established in [23], the work [32] presents an interesting nonconvex quadratic programming formulation of RRS. It is shown that the global minimizer of the nonconvex formulation recovers the true corruption level (still not the ground-true rotations directly) when $p^2 q^2 = \Omega(\log n / n)$ under the RCM. Unfortunately, the work does not provide a concrete algorithm that provably finds a global minimizer of the nonconvex formulation. In [29], the authors introduced and analyzed a depth descent algorithm for recovering the underlying rotation matrices. In the context of the RCM, they showed asymptotic convergence of their algorithm to the underlying rotations without providing a specific rate. The result is achieved under the conditions that the algorithm is initialized near $\boldsymbol{X}^\star$, $q \ge \mathcal{O}(\log n / n)$, and $p \ge 1 - 1/(d(d-1)+2)$. The latter requirement translates to $p \ge 3/4$ for $\mathrm{SO}(2)$ and $p \ge 7/8$ for $\mathrm{SO}(3)$. It is important to note, however, that the primary focus of their research lies in the adversarial corruption setup rather than the RCM.

**Main contributions.** Towards tackling the RRS problem under the measurement model (1), we consider the following least-unsquared formulation, which was introduced in [39] as the initial step for applying the SDR method:

$$
\underset{\boldsymbol{X} \in \mathbb{R}^{nd \times d}}{\text{minimize}} \ f(\boldsymbol{X}) := \sum_{(i,j) \in \mathcal{E}} \| \boldsymbol{X}_i \boldsymbol{X}_j^\top - \boldsymbol{Y}_{ij} \|_F
$$
$$
\text{subject to } \boldsymbol{X}_i \in \mathrm{SO}(d), \ 1 \le i \le n. \tag{2}
$$

Note that this problem is *nonsmooth* and *nonconvex* due to the unsquared Frobenius-norm loss and the rotation group constraint, respectively. We design a ***Riemannian Subgradient synchronization*** algorithm (ReSync) for addressing problem (2); see Algorithm 1. ReSync will first call an initialization procedure named SpectrIn (see Algorithm 2), which is a spectral relaxation method. Then, it implements an iterative Riemannian subgradient procedure. ReSync targets at directly recovering the ground-truth rotations $\boldsymbol{X}^\star \in \mathrm{SO}(d)^n$ rather than the Gram matrix or the corruption level. Under the RCM (see Section 2.2), we provide the following strong theoretical guarantees for ReSync:

(S.1) *Initialization.* The first step of ReSync is to call SpectrIn for computing the initial point $\boldsymbol{X}^0$. Theoretically, we establish that $\boldsymbol{X}^0$ can be relatively close to $\boldsymbol{X}^\star$ depending on $p$ and $q$; see Theorem 2.

(S.2) *Weak sharpness.* We then establish a problem-intrinsic property of the formulation (2) called weak sharpness; see Theorem 3. This property characterizes the geometry of problem (2) and is of independent interest.

(S.3) *Convergence analysis.* Finally, we derive the local linear rate of convergence for ReSync based on the established weak sharpness property; see Theorem 4.

The main idea is that the weak sharpness property in (S.2) helps to show *linear convergence* of ReSync to $\boldsymbol{X}^\star$ in (S.3). However, this result only holds *locally*. Thus, we need the initialization guarantee in (S.1) to initialize our algorithm in this local region and then argue that it will not leave this region once initialized. We refer to Sections 3.1 to 3.3 for more technical challenges and our proof ideas. Combining the above theoretical results yields our overall guarantee: ReSync *converges linearly to the ground-truth rotations $\boldsymbol{X}^\star$ when $p^7 q^2 = \Omega(\log n/n)$*; see Theorem 1.

**Notation.** Our notation is mostly standard. We use $\mathbb{R}^{nd \times d} \ni \boldsymbol{X} = (\boldsymbol{X}_1; \ldots; \boldsymbol{X}_n) \in \mathrm{SO}(d)^n$ to represent the Cartesian product of all the variables $\boldsymbol{X}_i \in \mathrm{SO}(d), 1 \le i \le n$. The same applies to the ground-truth rotations $\boldsymbol{X}^\star = (\boldsymbol{X}_1^\star; \cdots; \boldsymbol{X}_n^\star)$. Let $\mathcal{E}_i = \{j \mid (i,j) \in \mathcal{E}\}$, $\mathcal{A}_i = \{j \mid (i,j) \in \mathcal{A}\}$, and $\mathcal{A}_i^c = \mathcal{E}_i \setminus \mathcal{A}_i$. We also define $\mathcal{A}_{ij} = \mathcal{A}_i \cap \mathcal{A}_j$ for simplicity. For a set $S$, we use $|S|$ to denote its cardinality. For any matrix $\boldsymbol{X}, \boldsymbol{Y} \in \mathbb{R}^{nd \times d}$, we define the following distance up to a global rotation:

$$\mathrm{dist}\,(\boldsymbol{X}, \boldsymbol{Y}) = \|\boldsymbol{X} - \boldsymbol{Y}\boldsymbol{R}^\star\|_F, \text{ where } \boldsymbol{R}^\star = \underset{\boldsymbol{R} \in \mathrm{SO}(d)}{\arg\min} \|\boldsymbol{X}\boldsymbol{R} - \boldsymbol{Y}\|_F^2 = \mathcal{P}_{\mathrm{SO}(d)}(\boldsymbol{X}^\top \boldsymbol{Y}).$$

Besides, we introduce the following distances up to the global rotation $\boldsymbol{R}^\star$ defined above:

$$\mathrm{dist}_1\,(\boldsymbol{X}, \boldsymbol{Y}) = \sum_{i=1}^{n} \|\boldsymbol{X}_i - \boldsymbol{Y}_i \boldsymbol{R}^\star\|_F, \quad \mathrm{dist}_\infty\,(\boldsymbol{X}, \boldsymbol{Y}) = \max_{1 \le i \le n} \|\boldsymbol{X}_i - \boldsymbol{Y}_i \boldsymbol{R}^\star\|_F.$$

## 2 Algorithm and Setup

### 2.1 ReSync: Algorithm Development

In this subsection, we present ReSync for tackling the nonsmooth nonconvex formulation (2); see Algorithm 1. Our algorithm has two main parts, i.e., initialization and an iterative Riemannian subgradient procedure.

**Initialization.** ReSync first calls a procedure SpectrIn (see Algorithm 2) for initialization. SpectrIn is a spectral relaxation-based initialization technique. SpectrIn computes the first $d$ leading unit eigenvectors of the data matrix to form $\boldsymbol{\Phi} \in \mathbb{R}^{nd \times d}$. We multiply $\sqrt{n}$ to those eigenvectors to ensure that its norm matches that of $\mathrm{SO}(d)^n$. We also construct $\boldsymbol{\Psi}$, which reverses the sign of the last column of $\boldsymbol{\Phi}$ so that the determinants of $\boldsymbol{\Phi}$ and $\boldsymbol{\Psi}$ differ by a sign. Then, we compute the projection of $\boldsymbol{\Phi}$ and $\boldsymbol{\Psi}$ onto $\mathrm{SO}(d)^n$. The projection is computed in a block-wise manner, namely

$$\widetilde{\boldsymbol{\Phi}}_i = \mathcal{P}_{\mathrm{SO}(d)}(\boldsymbol{\Phi}_i), \quad 1 \le i \le n,$$

where $\boldsymbol{\Phi}_i, \widetilde{\boldsymbol{\Phi}}_i \in \mathbb{R}^{d \times d}$ are the $i$-th block of $\boldsymbol{\Phi}$ and $\widetilde{\boldsymbol{\Phi}}$, respectively. The projection can be explicitly evaluated as

$$\widetilde{\boldsymbol{\Phi}}_i = \begin{cases} \boldsymbol{P}_i \boldsymbol{Q}_i^\top, & \text{if } \det(\boldsymbol{\Phi}_i) > 0, \\ \widehat{\boldsymbol{P}}_i \boldsymbol{Q}_i^\top, & \text{otherwise,} \end{cases} \quad 1 \le i \le n.$$

Here, $\boldsymbol{P}_i, \boldsymbol{Q}_i \in \mathbb{R}^{d \times d}$ are the left and right singular vectors of $\boldsymbol{\Phi}_i$ (with descending order of singular values), respectively, and $\widehat{\boldsymbol{P}}_i$ is obtained by reversing the sign of the last column of $\boldsymbol{P}_i$. The initial point $\boldsymbol{X}^0$ is chosen as $\widetilde{\boldsymbol{\Phi}}$ or $\widetilde{\boldsymbol{\Psi}}$, depending on which is closer to $\mathrm{SO}(d)^n$.

Let us mention that the computation of $\widetilde{\boldsymbol{\Psi}}$ and Steps 5 - 9 in SpectrIn can practically improve the approximation error $\mathrm{dist}(\boldsymbol{X}^0, \boldsymbol{X}^\star)$. We demonstrate such a phenomenon in Figure 1, in which "Naive SpectrIn" refers to outputting $\boldsymbol{X}^0 = \widetilde{\boldsymbol{\Phi}}$ directly in Algorithm 2.

Figure 1: The average under 100 simulations of the initial distance $\mathrm{dist}(\boldsymbol{X}^0, \boldsymbol{X}^\star)$ computed by Algorithm 2 versus naive spectral initialization (i.e., outputting $\boldsymbol{X}^0 = \widetilde{\boldsymbol{\Phi}}$ directly) with $p = 0.2, q = 0.2$ and $d = 3$.

**Riemannian subgradient update.** ReSync then implements an iterative Riemannian subgradient procedure after obtaining the initial point $\boldsymbol{X}^0$. The key is to compute the search direction (Riemannian subgradient) $\widetilde{\nabla}_{\mathcal{R}} f(\boldsymbol{X}_i^k)$ and the retraction $\mathrm{Retr}_{\boldsymbol{X}_i^k}(\cdot)$ onto $\mathrm{SO}(d)$ for $1 \leq i \leq n$. Towards providing concrete formulas for the Riemannian subgradient update, let us impose the Euclidean inner product $\langle \boldsymbol{A}, \boldsymbol{B} \rangle = \mathrm{trace}(\boldsymbol{A}^\top \boldsymbol{B})$ as the inherent Riemannian metric. Consequently, the tangent space to $\mathrm{SO}(d)$ at $\boldsymbol{R} \in \mathrm{SO}(d)$ is given by $\mathrm{T}_{\boldsymbol{R}} := \{\boldsymbol{R}\boldsymbol{S} : \boldsymbol{S} \in \mathbb{R}^{d \times d}, \boldsymbol{S} + \boldsymbol{S}^\top = 0\}$. The Riemannian subgradient $\widetilde{\nabla}_{\mathcal{R}} f(\boldsymbol{X}_i)$ can be computed as [40, Theorem 5.1]

$$\widetilde{\nabla}_{\mathcal{R}} f(\boldsymbol{X}_i) = \mathcal{P}_{\mathrm{T}_{\boldsymbol{X}_i}}(\widetilde{\nabla} f(\boldsymbol{X}_i)), \quad 1 \leq i \leq n, \tag{3}$$

where the projection can be computed as $\mathcal{P}_{\mathrm{T}_{\boldsymbol{X}_i}}(\boldsymbol{B}) = \boldsymbol{X}_i\left(\boldsymbol{X}_i^\top \boldsymbol{B} - \boldsymbol{B}^\top \boldsymbol{X}_i\right)/2$ for any $\boldsymbol{B} \in \mathbb{R}^{d \times d}$ and $\widetilde{\nabla} f(\boldsymbol{X}_i)$ is the Euclidean subgradient of $f$ with respect to the $i$-th block variable $\boldsymbol{X}_i$. Let us define $f_{i,j}(\boldsymbol{X}) := \|\boldsymbol{X}_i \boldsymbol{X}_j^\top - \boldsymbol{Y}_{ij}\|_F$. The Euclidean subdifferential $\partial f(\boldsymbol{X}_i)$ with respect to the block variable $\boldsymbol{X}_i$ is given by

$$\partial f(\boldsymbol{X}_i) = 2 \sum_{j:(i,j) \in \mathcal{E}} \partial f_{i,j}(\boldsymbol{X}_i), \quad \text{with} \quad \partial f_{i,j}(\boldsymbol{X}_i) = \begin{cases} \frac{\boldsymbol{X}_i - \boldsymbol{Y}_{ij}\boldsymbol{X}_j}{\|\boldsymbol{X}_i\boldsymbol{X}_j^\top - \boldsymbol{Y}_{ij}\|_F}, & \text{if } \|\boldsymbol{X}_i\boldsymbol{X}_j^\top - \boldsymbol{Y}_{ij}\|_F \neq 0, \\ \boldsymbol{V} \in \mathbb{R}^{d \times d}, \ \|\boldsymbol{V}\|_F \leq 1, & \text{otherwise.} \end{cases}$$

---

**Algorithm 1** ReSync: Riemannian Subgradient Synchronization

**Require:** Initialize $\boldsymbol{X}^0 = \mathsf{SpectrIn}(\boldsymbol{Y})$ (Algorithm 2), where $\boldsymbol{Y} \in \mathbb{R}^{nd \times nd}$ and its $(i,j)$-th block is $\boldsymbol{Y}_{i,j} \in \mathbb{R}^{d \times d}$;
1: Set iteration count $k = 0$;
2: **while** stopping criterion not met **do**
3:    Update the step size $\mu_k$;
4:    Riemannian subgradient update:

$$\boldsymbol{X}_i^{k+1} = \mathrm{Retr}_{\boldsymbol{X}_i^k}\left(-\mu_k \widetilde{\nabla}_{\mathcal{R}} f(\boldsymbol{X}_i^k)\right)$$

   for $1 \leq i \leq n$;
5:    Update iteration count $k = k + 1$;
6: **end while**

---

**Algorithm 2** SpectrIn: Spectral Initialization

1: **Input:** $\boldsymbol{Y} \in \mathbb{R}^{nd \times nd}$;
2: Compute the $d$ leading unit eigenvectors of $\boldsymbol{Y}$: $\{\boldsymbol{u}_1, \ldots, \boldsymbol{u}_d\}$;
3: Set $\boldsymbol{\Phi} = \sqrt{n}[\boldsymbol{u}_1, \boldsymbol{u}_2, \ldots, \boldsymbol{u}_d] \in \mathbb{R}^{nd \times d}$ and $\boldsymbol{\Psi} = \sqrt{n}[\boldsymbol{u}_1, \boldsymbol{u}_2, \ldots, \boldsymbol{u}_{d-1}, -\boldsymbol{u}_d]$;
4: Compute $\widetilde{\boldsymbol{\Phi}} = \mathcal{P}_{\mathrm{SO}(d)^n}(\boldsymbol{\Phi})$ and $\widetilde{\boldsymbol{\Psi}} = \mathcal{P}_{\mathrm{SO}(d)^n}(\boldsymbol{\Psi})$;
5: **if** $\|\widetilde{\boldsymbol{\Phi}} - \boldsymbol{\Phi}\|_F \leq \|\widetilde{\boldsymbol{\Psi}} - \boldsymbol{\Psi}\|_F$ **then**
6:    $\boldsymbol{X}^0 = \widetilde{\boldsymbol{\Phi}}$;
7: **else**
8:    $\boldsymbol{X}^0 = \widetilde{\boldsymbol{\Psi}}$;
9: **end if**
10: **Output:** Initial point $\boldsymbol{X}^0$.

---

Any element $\widetilde{\nabla} f(\boldsymbol{X}_i) \in \partial f(\boldsymbol{X}_i)$ is called a Euclidean subgradient. In ReSync, one can choose an arbitrary subgradient $\widetilde{\nabla} f(\boldsymbol{X}_i) \in \partial f(\boldsymbol{X}_i)$ at $\boldsymbol{X}_i$.

Mimicking the gradient method to update along the search direction $\widetilde{\nabla}_{\mathcal{R}} f(\boldsymbol{X}_i)$ provides a point $\boldsymbol{X}_i^+ = \boldsymbol{X}_i - \mu \widetilde{\nabla}_{\mathcal{R}} f(\boldsymbol{X}_i)$ on the tangent space $\mathrm{T}_{\boldsymbol{X}_i}$ at $\boldsymbol{X}_i$, which may violate the manifold constraint "$\boldsymbol{X}_i^+ \in \mathrm{SO}(d)$". One common approach in Riemannian optimization is to employ a retraction operator to address the feasibility issue. For $\mathrm{SO}(d)$, we can use a QR decomposition-based retraction and implement the Riemannian subgradient step as

$$\boldsymbol{X}_i^+ = \mathrm{Retr}_{\boldsymbol{X}_i}\left(-\mu \widetilde{\nabla}_{\mathcal{R}} f(\boldsymbol{X}_i)\right) = \mathrm{Qr}\left(\boldsymbol{X}_i - \mu \widetilde{\nabla}_{\mathcal{R}} f(\boldsymbol{X}_i)\right), \quad 1 \leq i \leq n. \tag{4}$$

Here, $\mathrm{Qr}(\boldsymbol{B})$ returns the Q-factor in the thin QR decomposition of $\boldsymbol{B}$, while the diagonal entries of the R-factor are restricted to be positive [7].

Finally, setting $\boldsymbol{X}_i = \boldsymbol{X}_i^k$, $\boldsymbol{X}_j = \boldsymbol{X}_j^k$ for all $j$ such that $(i,j) \in \mathcal{E}$, $\mu = \mu_k$ in (3) and (4) yields a concrete implementation of Step 4 in ReSync and leads to $\mathrm{SO}(d) \ni \boldsymbol{X}_i^{k+1} = \boldsymbol{X}_i^+$ for $1 \leq i \leq n$. This completes the description of one full iteration of ReSync. Note that the per-iteration complexity of the Riemannian subgradient procedure is $\mathcal{O}(n^2 q)$, and Algorithm 2 has computational cost $\mathcal{O}(n^3)$.

## 2.2 RCM Setup for Theoretical Analysis

We develop our theoretical analysis of ReSync by adopting the random corruption model (RCM). The RCM was previously used in many works to analyze the performance of various synchronization algorithms; see, e.g., [39, 19, 23, 32]. Specifically, we can represent our measurement model (1) on a graph $\mathcal{G}(\mathcal{V}, \mathcal{E})$, where $\mathcal{V}$ is a set of $n$ nodes representing $\{\boldsymbol{X}_1^\star, \cdots, \boldsymbol{X}_n^\star\}$ and $\mathcal{E}$ is a set of edges

containing all the available measurements $\{Y_{i,j}, (i,j) \in \mathcal{E}\}$. We assume that the graph $\mathcal{G}$ follows the well-known Erdös-Rényi model $\mathsf{G}(n,q)$, which implies that each edge $(i,j) \in \mathcal{E}$ is observed with probability $q$, independently from every other edge. Each edge $(i,j) \in \mathcal{E}$ is a true observation (i.e., $(i,j) \in \mathcal{A}$) with probability $p$ and an outlier (i.e., $(i,j) \in \mathcal{A}^c$) with probability $1-p$. Furthermore, the outliers $\{O_{i,j}\}_{(i,j)\in\mathcal{A}^c}$ are assumed to be independently and uniformly distributed on $\mathrm{SO}(d)$.

## 3  Main Results

In this section, we present our theoretical results for ReSync. Our main results are summarized in the following theorem, which states that our proposed algorithm can converge at a linear rate to the underlying rotations $X^\star$. Our standing assumption in this section is stated below.

> All our theoretical results in this section are based on the RCM; see Section 2.2.

**Theorem 1** (overall). *Suppose that the ratios $p$ and $q$ satisfy*

$$p^7 q^2 = \Omega\left(\frac{\log n}{n}\right).$$

*With probability at least $1 - \mathcal{O}(1/n)$, ReSync with $\mu_k = \mu_0 \gamma^k$, where $\mu_0 = \Theta(p^2/n)$ and $\gamma = 1 - \frac{pq}{16}$, converges linearly to the ground-truth rotations $X^\star$ (up to a global rotation), i.e.,*

$$\mathrm{dist}\left(X^k, X^\star\right) \le \xi_0 \gamma^k, \quad \mathrm{dist}_\infty\left(X^k, X^\star\right) \le \delta_0 \gamma^k, \quad \forall k \ge 0.$$

*Here, $\xi_0 = \Theta(\sqrt{np^5 q})$ and $\delta_0 = \Theta(p^2)$.*

The basic idea of the proof is to establish the problem-intrinsic property of weak sharpness and then use it to derive a linear convergence result. However, the result only holds locally. Thus, we develop a procedure to initialize the algorithm in this local region and argue that ReSync will not leave this region afterwards. In the remaining parts of this section, we implement the above ideas and highlight the challenges and approaches to overcoming them.

### 3.1  Analysis of SpectrIn with Leave-One-Out Technique

**Theorem 2** (initialization). *Let $X^0$ be generated by SpectrIn (see Algorithm 2). Suppose that the ratios $p$ and $q$ satisfy*

$$p^2 q = \Omega\left(\frac{\log n}{n}\right).$$

*Then, with probability at least $1 - \mathcal{O}(1/n)$, we have*

$$\mathrm{dist}(X^0, X^\star) = \mathcal{O}\left(\frac{\sqrt{\log n}}{p\sqrt{q}}\right) \quad \text{and} \quad \mathrm{dist}_\infty(X^0, X^\star) = \mathcal{O}\left(\frac{\sqrt{\log n}}{p\sqrt{nq}}\right). \tag{5}$$

The works [34] and [11] show that exact reconstruction of $X^\star$ is information-theoretically possible if the condition $p^2 q = \Omega(\log n/n)$ holds for the cases $d = 2$ and $d = 3$, respectively. Though Theorem 2 does not provide exact recovery, it achieves an optimal sample complexity for reconstructing an approximate solution in the infinity norm. Specifically, Theorem 2 shows that, as long as $p^2 q \ge C \log n/n$ for some constant $C > 0$ large enough, the $\ell_\infty$-distance $\mathrm{dist}_\infty(X^0, X^\star)$ (i.e., $\max_{1 \le i \le n} \mathrm{dist}(X_i, X_i^\star)$) can be made relatively small. However, the $\ell_2$-distance $\mathrm{dist}(X^0, X^\star)$ is of the order $\Omega(\sqrt{n})$ under such a sample complexity.

The work [26] considers orthogonal and permutation group synchronization and shows that spectral relaxation-based methods achieve near-optimal performance bounds. Our result differs from that of [26] in twofold: 1) Our approach follows the standard leave-one-out analysis based on the standard "Dist" (up to $\mathrm{O}(d)$ invariance) defined above Lemma 3 in the Appendix. Nonetheless, we have to transfer the results to "dist" due to the structure of $\mathrm{SO}(d)$ in Lemma 5, which is a nontrivial step due to the specific structure of $\mathrm{SO}(d)$. 2) Our result can handle incomplete observations (i.e., $q < 1$). In the case of incomplete observations, the construction in (17) in the Appendix becomes more intricate; it has the additional third column, rendering the analysis of our Lemma 2 more involved.

We prove Theorem 2 with some matrix concentration bounds and the leave-one-out technique. We provide the proof sketch below and refer to Appendix A for the full derivations.

**Proof outline of Theorem 2.** According to (1) and the fact $\mathbb{E}(\boldsymbol{O}_{ij}) = \boldsymbol{0}$ since outliers are assumed to be independently and uniformly distributed on $\mathrm{SO}(d)$ in the RCM (see Appendix A), we know that $\mathbb{E}(\boldsymbol{Y}_{ij}) = pq\boldsymbol{X}_i^\star \boldsymbol{X}_j^{\star\top}$ for all $(i, j) \in [n] \times [n]$. This motivates us to introduce the noise matrix $\boldsymbol{W}_{ij} = \boldsymbol{Y}_{ij} - pq\boldsymbol{X}_i^\star \boldsymbol{X}_j^{\star\top}$, i.e.,

$$\boldsymbol{Y} = pq\boldsymbol{X}^\star \boldsymbol{X}^{\star\top} + \boldsymbol{W}. \tag{6}$$

The condition $p^2 q = \Omega(\log n/n)$ in Theorem 2 ensures that the expectation $pq\boldsymbol{X}^\star \boldsymbol{X}^{\star\top}$ will dominate the noise matrix $\boldsymbol{W}$ in the decomposition (6).

We first discuss how to bound $\mathrm{dist}(\boldsymbol{X}^0, \boldsymbol{X}^\star)$. Notice that $\boldsymbol{X}^0$ and $\boldsymbol{X}^\star$ are the $d$ leading eigenvectors of $\boldsymbol{Y}$ (after projection onto $\mathrm{SO}(d)^n$) and $pq\boldsymbol{X}^\star \boldsymbol{X}^{\star\top}$, respectively. We can then use the matrix perturbation theory (see Lemma 3) to bound $\mathrm{dist}(\boldsymbol{X}^0, \boldsymbol{X}^\star)$. Towards this end, we need to estimate the operator norm $\|\boldsymbol{W}\|_2$, which could be done by applying the standard matrix Bernstein concentration inequality [38] since the blocks $\{\boldsymbol{W}_{ij}\}$ are i.i.d. white noise with bounded operator norms and variances; see Lemma 2.

We next turn to bound the initialization error in the infinity norm, i.e., $\mathrm{dist}_\infty(\boldsymbol{X}^0, \boldsymbol{X}^\star)$. Let us use $(\boldsymbol{W}\boldsymbol{X}^0)_m \in \mathbb{R}^{d \times d}$ to denote the $m$-th block of $\boldsymbol{W}\boldsymbol{X}^0 \in \mathbb{R}^{nd \times d}$ for $1 \leq m \leq n$. The main technical challenge lies in deriving a sharp bound for the term $\max_{1 \leq m \leq n} \|(\boldsymbol{W}\boldsymbol{X}^0)_m\|_F$, as it involves two *dependent* random quantities, i.e., the noise matrix $\boldsymbol{W}$ and the initial $\boldsymbol{X}^0$ that is obtained by projecting the first $d$ leading eigenvectors of $\boldsymbol{Y}$ onto $\mathrm{SO}(d)^n$. To overcome such a statistical dependence, we utilize the leave-one-out technique. This technique was utilized in [42] to analyze the phase synchronization problem and was later applied to many other synchronization problems [1, 10, 15, 18, 26]. Let us define

$$\boldsymbol{Y}^{(m)} = pq\boldsymbol{X}^\star \boldsymbol{X}^{\star\top} + \boldsymbol{W}^{(m)} \quad \text{with} \quad \boldsymbol{W}_{kl}^{(m)} = \boldsymbol{W}_{kl} \cdot \mathbf{1}_{\{k \neq m\}} \cdot \mathbf{1}_{\{l \neq m\}}. \tag{7}$$

That is, we construct $\boldsymbol{W}^{(m)} \in \mathbb{R}^{nd \times nd}$ by setting the $m$-th block-wise row and column of $\boldsymbol{W}$ to be $\boldsymbol{0}$. Then, it is easy to see that $\boldsymbol{Y}^{(m)}$ is statistically independent of $\boldsymbol{W}_m^\top \in \mathbb{R}^{d \times nd}$, where the latter denotes the $m$-th block-wise row of $\boldsymbol{W}$. Let $\boldsymbol{X}^{(m)}$ be the $d$ leading eigenvectors of $\boldsymbol{Y}^{(m)}$. Consequently, $\boldsymbol{X}^{(m)}$ is also independent of $\boldsymbol{W}_m^\top$. Based on the above discussions, we can bound each $\|(\boldsymbol{W}\boldsymbol{X}^0)_m\|_F$ in the following way:

$$\|(\boldsymbol{W}\boldsymbol{X}^0)_m\|_F = \|\boldsymbol{W}_m^\top \boldsymbol{X}^0\|_F \leq \|\boldsymbol{W}_m^\top \boldsymbol{X}^{(m)}\|_F + \|\boldsymbol{W}_m^\top (\boldsymbol{X}^0 - \boldsymbol{X}^{(m)})\|_F. \tag{8}$$

The first term $\|\boldsymbol{W}_m^\top \boldsymbol{X}^{(m)}\|_F$ can be bounded using an appropriate concentration inequality due to the statistical independence between $\boldsymbol{W}_m^\top$ and $\boldsymbol{X}^{(m)}$. The second term can be bounded as

$$\|\boldsymbol{W}_m^\top (\boldsymbol{X}^0 - \boldsymbol{X}^{(m)})\|_F \leq \|\boldsymbol{W}_m\|_2 \cdot \|\boldsymbol{X}^0 - \boldsymbol{X}^{(m)}\|_F,$$

in which $\|\boldsymbol{W}_m\|_2$ can be further bounded by matrix concentration inequality (see Lemma 2) and $\|\boldsymbol{X}^0 - \boldsymbol{X}^{(m)}\|_F$ can be bounded using standard matrix perturbation theory (see Lemma 4).

## 3.2 Weak Sharpness and Exact Recovery

We next present a property that is intrinsic to problem (2) in the following theorem.

**Theorem 3** (weak sharpness). *Suppose that the ratios $p$ and $q$ satisfy*

$$p^2 q^2 = \Omega\left(\frac{\log n}{n}\right).$$

*Then, with probability at least $1 - \mathcal{O}(1/n)$, for any $\boldsymbol{X} \in \mathrm{SO}(d)^n$ satisfying $\mathrm{dist}_\infty(\boldsymbol{X}, \boldsymbol{X}^\star) = \mathcal{O}(p)$, we have*

$$f(\boldsymbol{X}) - f(\boldsymbol{X}^\star) \geq \frac{npq}{8} \, \mathrm{dist}_1(\boldsymbol{X}, \boldsymbol{X}^\star).$$

Some remarks on Theorem 3 are in order. This theorem shows that problem (2) possesses the *weak sharpness* property [8], which is intrinsic to the problem and independent of the algorithm used to solve it. It is known that with this property, various subgradient-type methods can achieve linear convergence [14, 25]. We will establish a similar linear convergence result for ReSync in the next subsection based on Theorem 3.

The weak sharpness property shown in Theorem 3 is of independent interest, as it could be helpful when analyzing other optimization algorithms (not just ReSync) for solving problem (2). Currently,

only a few applications are known to produce sharp optimization problems, such as robust low-rank matrix recovery [25], robust phase retrieval [16], and robust subspace recovery [24]. Furthermore, sharp instances of manifold optimization problems are especially scarce. Hence, Theorem 3 extends the list of optimization problems that possess the weak sharpness property and contributes to the growing literature on the geometry of structured nonsmooth nonconvex optimization problems.

It is worth noting that Theorem 3 also establishes the *exact recovery* property of the formulation (2). Specifically, up to a global rotation, the ground-truth $X^\star$ is guaranteed to be the unique global minimizer of $f$ over the region $\mathrm{SO}(d)^n \cap \{X : \mathrm{dist}_\infty(X, X^\star) = \mathcal{O}(p)\}$. Consequently, recovering the underlying $X^\star$ reduces to finding the global minimizer of $f$ over the aforementioned region. As we will show in the next subsection, ReSync will converge linearly to the global minimizer $X^\star$ when initialized in this region. However, the initialization requirement is subject to the stronger condition $p^4 q = \Omega\left(\log n / n\right)$ on the ratios $p$ and $q$, which is ensured by Theorem 2.

We list our main ideas for proving Theorem 3 below. The full proof can be found in Appendix B.

**Proof outline of Theorem 3.** Note that the objective function $f$ can be decomposed into two parts:

$$f(X) = \overbrace{\sum_{(i,j)\in\mathcal{A}} \|X_i^\top X_j - X_i^{\star\top} X_j^\star\|_F}^{g(X)} + \overbrace{\sum_{(i,j)\in\mathcal{A}^c} \|X_i^\top X_j - O_{ij}\|_F}^{h(X)}. \tag{9}$$

It is easy to see that $g(X^\star) = 0$ and $g(X) \geq 0$. Based on the fact that the true observation is uniformly distributed in all the indices, we have $\mathbb{E}\left(g(X)\right) = pq \sum_{1\leq i,j\leq n} \|X_i^\top X_j - X_i^{\star\top} X_j^\star\|_F \geq \frac{npq}{2}\mathrm{dist}_1(X, X^\star)$; see Appendix B.2 for the last inequality. A traditional way to lower bounding $g(X)$ using $\mathbb{E}(g(X))$ for all $X \in \mathrm{SO}(d)$ is to apply concentration inequality and an epsilon-net covering argument. Unfortunately, the sample complexity condition $p^2 q^2 = \Omega(\log n / n)$ does not lead to a high probability result in this way. Instead, our approach is to apply the concentration theory on the cardinalities of index sets rather than on $X$ directly; see the following lemma.

**Lemma 1** (concentration of cardinalities of index sets). *Given any $\epsilon = \Omega\left(\frac{\sqrt{\log n}}{\sqrt{npq}}\right)$, with probability at least $1 - \mathcal{O}(1/n)$, we have*

$$(1 - \epsilon)nq \leq |\mathcal{E}_i| \leq (1 + \epsilon)nq, \qquad\qquad (1 - \epsilon)npq \leq |\mathcal{A}_i| \leq (1 + \epsilon)npq,$$
$$(1 - \epsilon)npq^2 \leq |\mathcal{E}_i \cap \mathcal{A}_j| \leq (1 + \epsilon)npq^2, \qquad (1 - \epsilon)np^2q^2 \leq |\mathcal{A}_{ij}| \leq (1 + \epsilon)np^2q^2$$

*for any $1 \leq i, j \leq n$. See Section 1 for the notation.*

We then provide a sharp lower bound on $g(X)$ based on Lemma 1.

**Proposition 1.** *Under the conditions of Theorem 3, with probability at least $1 - \mathcal{O}(1/n)$, we have*

$$g(X) \geq \frac{3npq}{16} \mathrm{dist}_1\left(X, X^\star\right), \quad \forall X \in \mathrm{SO}(d)^n. \tag{10}$$

Next, to lower bound $h(X) - h(X^\star) = \sum_{(i,j)\in\mathcal{A}^c}\left(\|X_i^\top X_j - O_{ij}\|_F - \|X_i^{\star\top} X_j^\star - O_{ij}\|_F\right)$ we first bound

$$h(X) - h(X^\star) \geq \sum_{(i,j)\in\mathcal{A}^c} \left\langle \frac{X_i^{\star\top} X_j^\star - O_{ij}}{\|X_i^{\star\top} X_j^\star - O_{ij}\|_F}, X_i^\top X_j - X_i^{\star\top} X_j^\star \right\rangle,$$

where the inequality comes from the convexity of the norm function $U \mapsto \|U\|_F$ whenever $X_i^{\star\top} X_j^\star - O_{ij} \neq 0$. Then, using the orthogonality of each block of $X^\star$, we further have

$$h(X) - h(X^\star) \geq \sum_{(i,j)\in\mathcal{A}^c} \left\langle \frac{I - X_i^\star O_{ij} X_j^{\star\top}}{\|I - X_i^\star O_{ij} X_j^{\star\top}\|_F}, X_i^\star X_i^\top X_j X_j^{\star\top} - I \right\rangle. \tag{11}$$

Recall that since the outliers $\{O_{i,j}\}_{(i,j)\in\mathcal{A}^c}$ are independently and uniformly distributed on $\mathrm{SO}(d)$, so are $\{X_i^\star O_{ij} X_j^{\star\top}\}_{(i,j)\in\mathcal{A}^c}$. This observation indicates that $\left\{I - X_i^\star O_{ij} X_j^{\star\top}/\|I - X_i^\star O_{ij} X_j^{\star\top}\|_F\right\}_{(i,j)\in\mathcal{A}^c}$ are i.i.d. random matrices. Hence, by invoking concentration results that utilize the randomness of the outliers $\{O_{i,j}\}_{(i,j)\in\mathcal{A}^c}$ and the cardinalities $(i, j) \in \mathcal{A}^c$, we obtain the following result.

**Proposition 2.** *Under the conditions of Theorem 3, with probability at least $1 - \mathcal{O}(1/n)$, we have*

$$h(\boldsymbol{X}) - h(\boldsymbol{X}^\star) \geq -\frac{npq}{16} \operatorname{dist}_1(\boldsymbol{X}, \boldsymbol{X}^\star) \tag{12}$$

*for all $\boldsymbol{X} \in \mathrm{SO}(d)^n$ satisfying $\operatorname{dist}_\infty(\boldsymbol{X}, \boldsymbol{X}^\star) = \mathcal{O}(p)$.*

Combining Proposition 1 and Proposition 2 gives Theorem 3.

### 3.3 Convergence Analysis and Proof of Theorem 1

Let us now turn to utilize the weak sharpness property shown in Theorem 3 to establish the local linear convergence of ReSync. As a quick corollary of Theorem 3, we have the following result.

**Corollary 1.** *Under the conditions of Theorem 3, with probability at least $1 - \mathcal{O}(1/n)$, for any $\boldsymbol{X} \in \mathrm{SO}(d)^n$ satisfying $\operatorname{dist}_\infty(\boldsymbol{X}, \boldsymbol{X}^\star) = \mathcal{O}(p)$, we have*

$$\left\langle \widetilde{\nabla}_{\mathcal{R}} f(\boldsymbol{X}), \boldsymbol{X}^\star - \boldsymbol{X} \right\rangle \leq -\frac{npq}{16} \operatorname{dist}_1(\boldsymbol{X}, \boldsymbol{X}^\star), \quad \forall\, \widetilde{\nabla}_{\mathcal{R}} f(\boldsymbol{X}) \in \partial_{\mathcal{R}} f(\boldsymbol{X}). \tag{13}$$

This condition indicates that any Riemannian subgradient $\widetilde{\nabla}_{\mathcal{R}} f(\boldsymbol{X})$ provides a descent direction pointing towards $\boldsymbol{X}^\star$. However, it only holds for $\boldsymbol{X} \in \mathrm{SO}(d)^n$ satisfying $\operatorname{dist}_\infty(\boldsymbol{X}, \boldsymbol{X}^\star) = \mathcal{O}(p)$. Our key idea for establishing local convergence is to show that the Riemannian subgradient update in ReSync is a contraction operator *in both the Euclidean and infinity norm-induced distances* using Corollary 1, i.e., if $\boldsymbol{X}^k$ lies in the local region, then $\boldsymbol{X}^{k+1}$ also lies in the region. This idea motivates us to define two sequences of neighborhoods as follows:

$$\mathcal{N}_F^k = \{\boldsymbol{X} \mid \operatorname{dist}(\boldsymbol{X}, \boldsymbol{X}^\star) \leq \xi_k\} \quad \text{and} \quad \mathcal{N}_\infty^k = \{\boldsymbol{X} \mid \operatorname{dist}_\infty(\boldsymbol{X}, \boldsymbol{X}^\star) \leq \delta_k\}. \tag{14}$$

Here, $\xi_k = \xi_0 \gamma^k, \delta_k = \delta_0 \gamma^k$, where $\xi_0, \delta_0$, and $\gamma \in (0, 1)$ will be specified later. Thus, these two sequences of sets $\{\mathcal{N}_F^k\}$ and $\{\mathcal{N}_\infty^k\}$ will linearly shrink to the ground-truth. It remains to show that if $\boldsymbol{X}^k \in \mathcal{N}_F^k \cap \mathcal{N}_\infty^k$, then $\boldsymbol{X}^{k+1} \in \mathcal{N}_F^{k+1} \cap \mathcal{N}_\infty^{k+1}$, which is summarized in the following theorem.

**Theorem 4** (convergence analysis). *Suppose that $\delta_0 = \mathcal{O}(p^2)$ and $\xi_0 = \mathcal{O}(\sqrt{npq}\delta_0)$. Set $\gamma = 1 - \frac{pq}{16}$ and $\mu_k = \delta_k/n$ in ReSync. If $\boldsymbol{X}^k \in \mathcal{N}_F^k \cap \mathcal{N}_\infty^k$ for any $k \geq 0$, then with probability at least $1 - \mathcal{O}(1/n)$, we have*

$$\boldsymbol{X}^{k+1} \in \mathcal{N}_F^{k+1} \cap \mathcal{N}_\infty^{k+1}.$$

**Proof outline of Theorem 4.** The proof consisted of two parts. On the one hand, we need to show that $\boldsymbol{X}^{k+1} \in \mathcal{N}_F^{k+1}$, which can be achieved by applying Corollary 1. On the other hand, in order to show that $\boldsymbol{X}^{k+1} \in \mathcal{N}_\infty^{k+1}$, we need a good estimate of each block of $\widetilde{\nabla}_{\mathcal{R}} f(\boldsymbol{X})$. See Appendix C.

Having developed the necessary tools, we are now ready to prove Theorem 1.

**Proof of Theorem 1.** Based on Theorem 2, we know that $\boldsymbol{X}^0 \in \mathcal{N}_F^0 \bigcap \mathcal{N}_\infty^0$ if $\xi_0$ and $\delta_0$ satisfy

$$\xi_0 = \mathcal{O}\left(\frac{\sqrt{\log n}}{p\sqrt{q}}\right) \quad \text{and} \quad \delta_0 = \mathcal{O}\left(\frac{\sqrt{\log n}}{p\sqrt{nq}}\right). \tag{15}$$

According to Theorem 4, by choosing $\delta_0 = \Theta(p^2)$ and $\xi_0 = \Theta(\sqrt{np^5q})$, condition (15) holds when $p^7 q^2 = \Omega(\log n/n)$. This completes the proof of Theorem 1.

## 4 Experiments

In this section, we conduct experiments on ReSync for solving the RRS problem on both synthetic and real data, providing empirical support for our theoretical findings. Our experiments are conducted on a personal computer with a 2.90GHz 8-core CPU and 32GB memory. All our experiment results are averaged over 20 independent trials. Our code is available at `https://github.com/Huikang2019/ReSync`.

### 4.1 Synthetic Data

We consider the rotation group $\mathrm{SO}(3)$ in all our experiments. We generate $\boldsymbol{X}_1^\star, \ldots, \boldsymbol{X}_n^\star$ by first generating matrices of the same dimension with i.i.d. standard Gaussian entries and then projecting

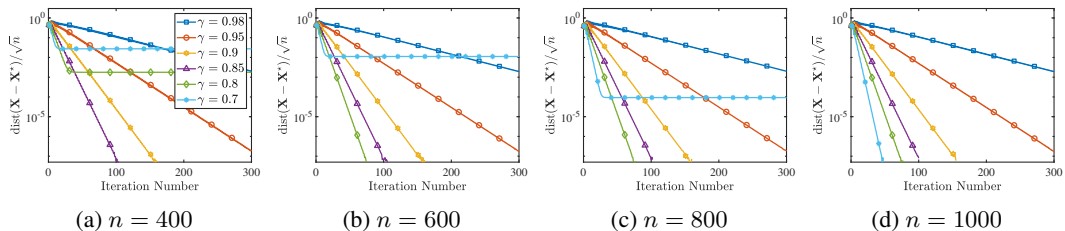

(a) $n = 400$      (b) $n = 600$      (c) $n = 800$      (d) $n = 1000$

Figure 2: Convergence of ReSync with $p = q = (\log n / n)^{1/3}$.

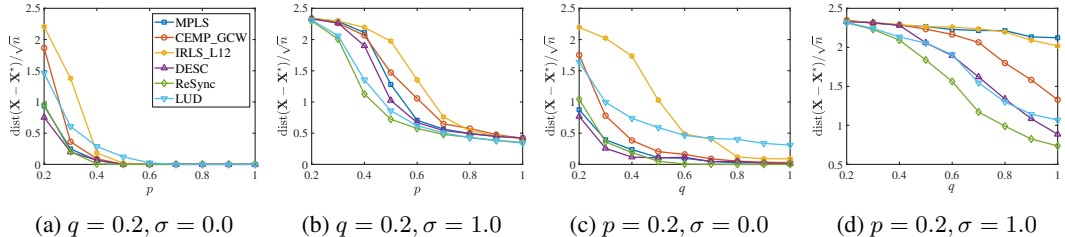

(a) $q = 0.2, \sigma = 0.0$    (b) $q = 0.2, \sigma = 1.0$    (c) $p = 0.2, \sigma = 0.0$    (d) $p = 0.2, \sigma = 1.0$

Figure 3: Comparison with state-of-the-art synchronization algorithms.

each of them onto $\mathrm{SO}(3)$. The underlying graph, outliers, and relative rotations in the measurement model (1) are generated according to the RCM as described in Section 2.2. In our experiments, we also consider the case where the true observations are contaminated by additive noise, namely, $\{\boldsymbol{Y}_{i,j}\}_{(i,j)\in\mathcal{A}}$ in (1) is generated using the formula

$$\boldsymbol{Y}_{i,j} = \mathcal{P}_{\mathrm{SO}(3)} \left( \boldsymbol{X}_i^{\star\top} \boldsymbol{X}_j^{\star} + \sigma \boldsymbol{G}_{i,j} \right) \quad \text{for} \quad (i,j) \in \mathcal{A}, \tag{16}$$

where $\boldsymbol{G}_{i,j}$ consists of i.i.d. entries following the standard Gaussian distribution and $\sigma \geq 0$ controls the variance level of the noise.

**Convergence verification of ReSync.** We evaluate the convergence performance of ReSync with the noise level $\sigma = 0$ in (16). We set $p = q = (\log n / n)^{1/3}$ in the measurement model (1), which satisfies $p^2 q = \log n / n$. We use the initial step size $\mu_0 = 1/npq$ and the decaying factor $\gamma \in \{0.7, 0.8, 0.85, 0.90, 0.95, 0.98\}$ in ReSync. We test the performance for various $n$ selected from $\{400, 600, 800, 1000\}$. Figure 2 displays the experiment results. It can be observed that (i) ReSync converges linearly to ground-truth rotations for a wide range of $\gamma$ and (ii) a smaller $\gamma$ often leads to faster convergence speed. These corroborate our theoretical findings. However, it is worth noting that excessively small $\gamma$ values may result in an early stopping phenomenon (e.g., $\gamma \leq 0.8$ when $n = 400$). In addition, ReSync performs better with a larger $n$, as it allows for a smaller $\gamma$ (e.g., $\gamma = 0.7$ when $n = 1000$) and hence converges to the ground-truth rotations faster.

**Comparison with the state-of-the-arts.** We next compare ReSync with state-of-the-art synchronization algorithms, including IRLS_L12 [9], MPLS [31], CEMP_GCW [23, 32], DESC [32], and LUD [39]. We obtain the implementation of the first four algorithms from `https://github.com/ColeWyeth/DESC`, while LUD's implementation is obtained through private communication with its authors. In our comparisons, we use their default parameter settings. For ReSync, we set the initial step size to $\mu_0 = 1/npq$ and the decaying factor to $\gamma = 0.95$, as suggested by the previous experiment. We fix $n = 200$ and vary the true observation ratio $p$ (or the observation ratio $q$) while keeping $q = 0.2$ (or $p = 0.2$) fixed. We display the experiment results for $\sigma = 0$ and $\sigma = 1$ in Figures 3a and 3b, respectively, where $p$ is selected from $\{0.2, 0.3, 0.4, \ldots, 1\}$. When $\sigma = 0$, ReSync achieves competitive performance compared to other robust synchronization algorithms. When the additive noise level is $\sigma = 1$, ReSync outperforms other algorithms. In Figures 3c and 3d, we present the results with varying $q$ chosen from $\{0.2, 0.3, 0.4, \ldots, 1\}$ for noise-free ($\sigma = 0$) and noisy ($\sigma = 1$) cases, respectively. In the noise-free case, DESC performs best when $q < 0.5$, while ReSync slightly outperforms others when $q \geq 0.5$. In the noisy case, it is clear that ReSync achieves the best performance for a large range of $q$.

## 4.2 Real Data

We consider the global alignment problem of three-dimensional scans from the Lucy dataset, which is a down-sampled version of the dataset containing 368 scans with a total number of 3.5 million triangles. We refer to [39] for more details about the experiment setting. We apply three algorithms LUD [39], DESC [32] and our ReSync on this dataset since they have the best performance on noisy synthetic data. As Figure 4 shows, ReSync outperforms the other two methods.

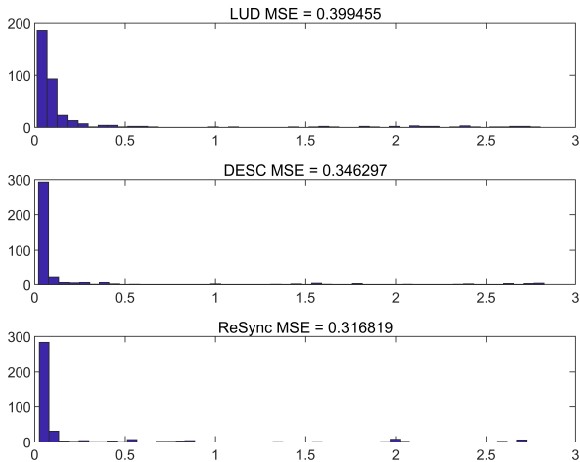

Figure 4: Histogram of the unsquared residuals of LUD, DESC, and ReSync for the Lucy dataset.

## 5 Conclusion and Discussions on Limitations

In this work, we introduced ReSync, a Riemannian subgradient-based algorithm with spectral initialization for solving RRS. We established strong theoretical results for ReSync under the RCM. In particular, we first presented an initialization guarantee for SpectrIn, which is a procedure embedded in ReSync for initialization. Then, we established a problem-intrinsic property called weak sharpness for our nonsmooth nonconvex formulation, which is of independent interest. Based on the established weak sharpness property, we derived linear convergence of ReSync to the underlying rotations once it is initialized in a local region. Combining these theoretical results demonstrates that ReSync converges linearly to the ground-truth rotations under the RCM.

**Limitations.** Our overall guarantee in Theorem 1 requires the sample complexity of $p^7 q^2 = \Omega(\log n/n)$, which does not match the currently best known lower bound $p^2 q = \Omega(\log n/n)$ for exact recovery [34, 11]. We showed in Theorem 2 that approximate recovery with an optimal sample complexity is possible. Moreover, we showed in Theorem 3 that exact recovery with $p^2 q^2 = \Omega(\log n/n)$ is possible if we have a global minimizer of the objective function of problem (2) within a certain local region. However, due to the nonconvexity of problem (2), it is non-trivial to obtain the said minimizer. We circumvented this difficulty by establishing the linear convergence of ReSync to a desired minimizer in Theorem 4. Nevertheless, a strong requirement on initialization is needed, which translates to the weaker final complexity result of $p^7 q^2 = \Omega(\log n/n)$.

Although our theory allows for $p \to 0$ as $n \to \infty$, our argument relies heavily on the randomness of the outliers $\{\boldsymbol{O}_{i,j}\}$ and the absence of additive noise. In practice, adversarial outliers that arbitrarily corrupt a measurement and additive noise contamination are prevalent. It remains unknown how well ReSync performs in such scenarios.

The above challenges are significant areas for future research and improvements.

## Acknowledgments and Disclosure of Funding

The authors thank Dr. Zengde Deng (Cainiao Network) and Dr. Shixiang Chen (University of Science and Technology of China) for providing helpful advice. They also thank the reviewers for their insightful comments, which have helped greatly to improve the quality and presentation of the manuscript.

Huikang Liu is supported in part by the National Natural Science Foundation of China (NSFC) Grant 72192832. Xiao Li is supported in part by the National Natural Science Foundation of China (NSFC) under grants 12201534 and 72150002, and in part by the Shenzhen Science and Technology Program under grants RCBS20210609103708017 and RCYX20221008093033010. Anthony Man-Cho So is supported in part by the Hong Kong Research Grants Council (RGC) General Research Fund (GRF) project CUHK 14205421.

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

# Contents

# A   Full Proof of Theorem 2

In this section, we present the full proof of Theorem 2. We will use the notation defined in the proof outline of Theorem 2 in Section 3.1.

Firstly, we know that

$$
\boldsymbol{W}_{ij} = \begin{cases} (1-pq)\boldsymbol{X}_i^{\star}\boldsymbol{X}_j^{\star\top}, & \text{with probablity } pq, \\ \boldsymbol{O}_{ij} - pq\boldsymbol{X}_i^{\star}\boldsymbol{X}_j^{\star\top}, & \text{with probablity } (1-p)q, \\ -pq\boldsymbol{X}_i^{\star}\boldsymbol{X}_j^{\star\top}, & \text{otherwise.} \end{cases} \tag{17}
$$

Since $\boldsymbol{O}_{ij}$ is assumed to be uniformly distributed on $\mathrm{SO}(d)$ in the RCM, given any matrix $\boldsymbol{Q} \in \mathrm{SO}(d)$, it is easy to see that $\boldsymbol{O}_{ij}\boldsymbol{Q}$ is also uniformly distributed on $\mathrm{SO}(d)$, so we have

$$
\mathbb{E}(\boldsymbol{O}_{ij}) = \mathbb{E}(\boldsymbol{O}_{ij}\boldsymbol{Q}) = \mathbb{E}(\boldsymbol{O}_{ij}) \cdot \boldsymbol{Q}, \quad \forall \boldsymbol{Q} \in \mathrm{SO}(d).
$$

Let $\boldsymbol{E}_{kl} \in \mathrm{SO}(d), k \neq l$ denote the diagonal matrix whose $k$-th and $l$-th diagonal entries are $-1$ and others are $1$, then we have $\mathbb{E}(\boldsymbol{O}_{ij}) = \mathbb{E}(\boldsymbol{O}_{ij}) \cdot \boldsymbol{E}_{kl}$, which implies

$$
d\mathbb{E}(\boldsymbol{O}_{ij}) = \mathbb{E}(\boldsymbol{O}_{ij}) \cdot (\boldsymbol{E}_{12} + \boldsymbol{E}_{23} + \cdots + \boldsymbol{E}_{d1}) = (d-4)\mathbb{E}(\boldsymbol{O}_{ij}).
$$

Thus, we have $\mathbb{E}(\boldsymbol{O}_{ij}) = \boldsymbol{0}$. Then, it is easy to see that $\mathbb{E}(\boldsymbol{W}_{ij}) = 0$ and

$$
\mathrm{Var}(\boldsymbol{W}_{ij}) = (1-pq)^2\boldsymbol{I} + (1-q)p^2q^2\boldsymbol{I} + (1-p)q(1+p^2q^2)\boldsymbol{I} = q(1-p^2q)\boldsymbol{I}. \tag{18}
$$

Based on the above calculations, we can derive the following Lemma, which is a direct result of the matrix Bernstein inequality [38]. Similar results can also be found in [42, Lemma 9], [27, Proposition 3], and [26, Eq. (5.12)].

**Lemma 2.** *With probability at least $1 - \mathcal{O}(1/n)$, the following holds for any $m \in [n]$:*

$$
\|\boldsymbol{W}\|_2 = \mathcal{O}\left(\sqrt{nq\log n}\right), \quad \|\boldsymbol{W}^{(m)}\|_2 = \mathcal{O}\left(\sqrt{nq\log n}\right), \quad \|\boldsymbol{W}_m\|_2 = \mathcal{O}\left(\sqrt{nq\log n}\right).
$$

*Proof.* Note that $\boldsymbol{W} = \sum_{i<j} \boldsymbol{W}^{(ij)}$, where $\boldsymbol{W}^{(ij)} \in \mathbb{R}^{nd \times nd}$ denotes the matrix with the $(i,j)$ and $(j,i)$-th block equal to $\boldsymbol{W}_{ij}$ and $\boldsymbol{W}_{ji}$ and others equal to $0$. So $\{\boldsymbol{W}^{(ij)}\}$ are i.i.d. centered and bounded random matrices. Besides, we have

$$
\left\|\mathbb{E}(\boldsymbol{W}\boldsymbol{W}^{\top})\right\|_2 = \left\|\sum_{i<j} \mathbb{E}(\boldsymbol{W}^{(ij)}(\boldsymbol{W}^{(ij)})^{\top})\right\|_2 = 2(n-1)\|\mathrm{Var}(\boldsymbol{W}_{ij})\|_2 = \mathcal{O}(nq).
$$

According to the matrix Bernstein inequality [38], we have that $\|\boldsymbol{W}\|_2 = \mathcal{O}\left(\sqrt{nq\log n}\right)$ holds with probability at least $1 - \mathcal{O}(1/n^2)$. The above argument also holds for each $\boldsymbol{W}^{(m)}$, then taking a union bound over the choice of $m \in [n]$ yields the second result. For $\boldsymbol{W}_m$, we have

$$
\|\boldsymbol{W}\|_2 = \max_{\|\boldsymbol{u}\|_2=1} \|\boldsymbol{W}\boldsymbol{u}\|_2 \geq \max_{\|\boldsymbol{u}\|_2=1} \|\boldsymbol{W}_m\boldsymbol{u}\|_2 = \|\boldsymbol{W}_m\|_2,
$$

which gives the last result. $\qquad\square$

The following lemma follows from [13]; see also [26, Theorem A.2]. [42, Lemma 11] is a special case where $d = 1$. Before that, we need to introduce the distance up to a global orthogonal matrix:

$$
\mathrm{Dist}\,(\boldsymbol{X},\boldsymbol{Y}) = \|\boldsymbol{X} - \boldsymbol{Y}\boldsymbol{R}^{\star}\|_F, \text{ where } \boldsymbol{R}^{\star} = \arg\min_{\boldsymbol{R} \in O(d)} \|\boldsymbol{X}\boldsymbol{R} - \boldsymbol{Y}\|_F^2 = \mathcal{P}_{O(d)}(\boldsymbol{X}^{\top}\boldsymbol{Y}).
$$

**Lemma 3** (Davis-Kahan $\sin\Theta$ Theorem). *Suppose that $\boldsymbol{A}, \boldsymbol{E} \in \mathcal{C}^{n \times n}$ are Hermitian matrices and $\tilde{\boldsymbol{A}} = \boldsymbol{A} + \boldsymbol{E}$. Let $\delta = \lambda_d(\boldsymbol{A}) - \lambda_{d+1}(\boldsymbol{A}) > 0$ be the gap between the $d$-th eigenvalue and $d+1$-th eigenvalue of $\boldsymbol{A}$ for some $1 \leq d \leq n - 1$. Furthermore, let $\boldsymbol{U}, \tilde{\boldsymbol{U}}$ be the $d$-leading eigenvectors of $\boldsymbol{A}$ and $\tilde{\boldsymbol{A}}$, respectively, which are normalized such that $\|\boldsymbol{U}\|_F = \|\tilde{\boldsymbol{U}}\|_F = \sqrt{nd}$. Then, we have*

$$
\mathrm{Dist}(\boldsymbol{U}, \tilde{\boldsymbol{U}}) \leq \frac{\sqrt{2}\|\boldsymbol{E}\boldsymbol{U}\|_F}{\delta - \|\boldsymbol{E}\|_2}. \tag{19}
$$

## A.1 Initialization Error in Euclidean Norm

Based on Lemma 2 and Lemma 3, we have the following bound on Dist. Note that the notations $\boldsymbol{\Phi}, \boldsymbol{\Psi}, \widetilde{\boldsymbol{\Phi}}, \widetilde{\boldsymbol{\Psi}}$ used in the following analysis are defined in SpectrIn (i.e., Algorithm 2).

**Lemma 4.** *Let $\boldsymbol{\Phi}$ and $\boldsymbol{\Phi}^{(m)}$ be the $d$-leading eigenvectors of $\boldsymbol{Y}$ and $\boldsymbol{Y}^{(m)}$, respectively, which are normalized such that $\|\boldsymbol{\Phi}\|_F = \|\boldsymbol{\Phi}^{(m)}\|_F = \sqrt{nd}$. Then, we have*

$$\mathrm{Dist}(\boldsymbol{\Phi}, \boldsymbol{X}^\star) = \mathcal{O}\left(\frac{\sqrt{\log n}}{p\sqrt{q}}\right) \quad \text{and} \quad \mathrm{Dist}(\boldsymbol{\Phi}, \boldsymbol{\Phi}^{(m)}) = \mathcal{O}(1) \tag{20}$$

*hold with probability at least $1 - \mathcal{O}(1/n)$.*

*Proof.* Let us Choose $\boldsymbol{A} = \boldsymbol{Y}^{(m)}$ and $\boldsymbol{E} = \Delta \boldsymbol{W}^{(m)} = \boldsymbol{W} - \boldsymbol{W}^{(m)}$ in Lemma 3, then $\tilde{\boldsymbol{A}} = \boldsymbol{Y}$, $\boldsymbol{U} = \boldsymbol{\Phi}^{(m)}$ and $\tilde{\boldsymbol{U}} = \boldsymbol{\Phi}$. Since $\boldsymbol{\Phi}^{(m)}$ is independent of $\Delta \boldsymbol{W}^{(m)}$, similar to Lemma 2, we apply the matrix Bernstein inequality [38] to obtain, with probability at least $1 - \mathcal{O}(1/n^2)$, that

$$\|\boldsymbol{EU}\|_F = \|\Delta \boldsymbol{W}^{(m)} \boldsymbol{\Phi}^{(m)}\|_F = \mathcal{O}(\sqrt{nq\log n}).$$

In addition, $\boldsymbol{Y}^{(m)} = pq\boldsymbol{X}^\star \boldsymbol{X}^{\star\top} + \boldsymbol{W}^{(m)}$ implies that

$$\delta = \lambda_d(\boldsymbol{Y}^{(m)}) - \lambda_{d+1}(\boldsymbol{Y}^{(m)}) \geq \lambda_d(pq\boldsymbol{X}^\star \boldsymbol{X}^{\star\top}) - \|\boldsymbol{W}^{(m)}\|_2 \geq npq - \mathcal{O}(\sqrt{nq\log n}).$$

where the second inequality holds due to $\lambda_d(\boldsymbol{X}^\star \boldsymbol{X}^{\star\top}) = n$ and the last inequality is from $\lambda_d(\boldsymbol{X}^\star \boldsymbol{X}^{\star\top}) = n$ and Lemma 2. Based on the condition that $p^2 q = \Omega\left(\frac{\log n}{n}\right)$, as long as $p^2 q \geq \frac{C\log n}{n}$ for some large enough constant $C$, we have

$$\delta - \|\boldsymbol{E}\|_2 \geq npq - \mathcal{O}(\sqrt{nq\log n}) - \|\boldsymbol{E}\|_2 \geq npq - \mathcal{O}(\sqrt{nq\log n}) \geq \frac{1}{2}npq,$$

where the second inequality holds because of $\|\boldsymbol{E}\|_2 \leq \|\boldsymbol{W}\|_2 + \|\boldsymbol{W}^{(m)}\|_2 = \mathcal{O}(\sqrt{nq\log n})$. Hence, by applying Lemma 3, we get

$$\mathrm{Dist}(\boldsymbol{\Phi}, \boldsymbol{\Phi}^{(m)}) \leq \frac{\sqrt{2}\|\boldsymbol{EU}\|_F}{\delta - \|\boldsymbol{E}\|_2} \leq \frac{\mathcal{O}(\sqrt{nq\log n})}{npq} = \mathcal{O}(1) \tag{21}$$

where the last inequality is because of $p\sqrt{q} = \Omega(\sqrt{\log n/n})$. Similarly, by choosing $\boldsymbol{A} = pq\boldsymbol{X}^\star \boldsymbol{X}^{\star\top}$ and $\boldsymbol{E} = \boldsymbol{W}$, we can show that

$$\mathrm{Dist}(\boldsymbol{\Phi}, \boldsymbol{X}^\star) \leq \frac{\sqrt{2}\|\boldsymbol{W}\boldsymbol{X}^\star\|_F}{npq - \mathcal{O}(\sqrt{nq\log n})} \leq \frac{\sqrt{2}\|\boldsymbol{W}\|_2 \|\boldsymbol{X}^\star\|_F}{\frac{1}{2}npq} = \mathcal{O}\left(\frac{\sqrt{\log n}}{p\sqrt{q}}\right).$$

Here, the last inequality holds because $\|\boldsymbol{W}\|_2 = \mathcal{O}(\sqrt{nq\log n})$ (see Lemma 2) and the fact that $\|\boldsymbol{X}^\star\|_F = \sqrt{nd}$. $\qquad \square$

Following the same analysis as in Lemma 4, we are also able to show that $\mathrm{Dist}(\boldsymbol{\Psi}, \boldsymbol{X}^\star) = \mathcal{O}\left(\frac{\sqrt{\log n}}{p\sqrt{q}}\right)$, where $\boldsymbol{\Psi}$ reverses the sign of the last column of $\boldsymbol{\Phi}$ so that the determinants of $\boldsymbol{\Phi}$ and $\boldsymbol{\Psi}$ differ by a sign. However, Lemma 4 only provides the upper bound on "Dist", i.e., the distance up to an orthogonal matrix. The following lemma translates the result to that on "dist".

**Lemma 5.** *Suppose that $\|\widetilde{\boldsymbol{\Phi}} - \boldsymbol{\Phi}\|_F \leq \|\widetilde{\boldsymbol{\Psi}} - \boldsymbol{\Psi}\|_F$, where $\widetilde{\boldsymbol{\Phi}} = \mathcal{P}_{\mathrm{SO}(d)^n}(\boldsymbol{\Phi})$ and $\widetilde{\boldsymbol{\Psi}} = \mathcal{P}_{\mathrm{SO}(d)^n}(\boldsymbol{\Psi})$, then we have*

$$\mathrm{dist}(\boldsymbol{\Phi}, \boldsymbol{X}^\star) = \mathrm{Dist}(\boldsymbol{\Phi}, \boldsymbol{X}^\star) = \mathcal{O}\left(\frac{\sqrt{\log n}}{p\sqrt{q}}\right).$$

*Proof.* Based on the structure of $O(d)$ and $\mathrm{SO}(d)$, it is easy to see that

$$\mathrm{Dist}(\boldsymbol{\Phi}, \boldsymbol{X}^\star) = \min\{\mathrm{dist}(\boldsymbol{\Phi}, \boldsymbol{X}^\star), \mathrm{dist}(\boldsymbol{\Psi}, \boldsymbol{X}^\star)\}.$$

Our remaining task is to prove $\mathrm{Dist}(\boldsymbol{\Phi}, \boldsymbol{X}^\star) = \mathrm{dist}(\boldsymbol{\Phi}, \boldsymbol{X}^\star)$ based on the condition that $\|\widetilde{\boldsymbol{\Phi}} - \boldsymbol{\Phi}\|_F \leq \|\widetilde{\boldsymbol{\Psi}} - \boldsymbol{\Psi}\|_F$. If $\mathrm{Dist}(\boldsymbol{\Phi}, \boldsymbol{X}^\star) = \mathrm{dist}(\boldsymbol{\Psi}, \boldsymbol{X}^\star)$, then we have

$$\mathcal{O}\left(\frac{\sqrt{\log n}}{p\sqrt{q}}\right) = \mathrm{Dist}(\boldsymbol{\Phi}, \boldsymbol{X}^\star) = \mathrm{dist}(\boldsymbol{\Psi}, \boldsymbol{X}^\star) \geq \|\widetilde{\boldsymbol{\Psi}} - \boldsymbol{\Psi}\|_F \geq \|\widetilde{\boldsymbol{\Phi}} - \boldsymbol{\Phi}\|_F$$

where the first inequality holds because $\widetilde{\boldsymbol{\Psi}}$ is the projection of $\boldsymbol{\Psi}$ on $\mathrm{SO}(d)^n$. It is easy to see that

$$\|\widetilde{\boldsymbol{\Phi}} - \boldsymbol{\Phi}\|_F + \|\widetilde{\boldsymbol{\Psi}} - \boldsymbol{\Psi}\|_F = \|\boldsymbol{\Phi} - \mathcal{P}_{\mathrm{SO}(d)^n}(\boldsymbol{\Phi})\|_F + \|\boldsymbol{\Phi} - \mathcal{P}_{(O(d)\backslash \mathrm{SO}(d))^n}(\boldsymbol{\Phi})\|_F \geq 2\sqrt{n}.$$

The equality holds because the mapping that reverses the sign of the last column of a matrix is a bijection between $\mathrm{SO}(d)$ and $O(d) \backslash \mathrm{SO}(d)$, and the inequality holds since the minimum distance between $\mathrm{SO}(d)$ and $O(d) \backslash \mathrm{SO}(d)$ is 2. Under the condition that $p^2 q = \Omega\left(\frac{\log n}{n}\right)$, as long as $p^2 q \geq \frac{C \log n}{n}$ for some large enough constant $C$, we could have $\|\widetilde{\boldsymbol{\Phi}} - \boldsymbol{\Phi}\|_F + \|\widetilde{\boldsymbol{\Psi}} - \boldsymbol{\Psi}\|_F = \mathcal{O}\left(\frac{\sqrt{\log n}}{p\sqrt{q}}\right) < 2\sqrt{n}$, which contradict to the above inequality. Thus, we have

$$\mathrm{dist}(\boldsymbol{\Phi}, \boldsymbol{X}^\star) = \mathrm{Dist}(\boldsymbol{\Phi}, \boldsymbol{X}^\star) = \mathcal{O}\left(\frac{\sqrt{\log n}}{p\sqrt{q}}\right).$$

$\square$

## A.2 Initialization Error in Infinity Norm

Next, based on the Davis-Kahan theorem, we can bound the distance between $\widetilde{\boldsymbol{X}}^0$ and $\boldsymbol{X}^\star$ in the infinity norm, which is stated in the following result.

**Lemma 6.** *Let $\boldsymbol{\Phi}$ be the $d$-leading eigenvectors of $\boldsymbol{Y}$. Then, we have*

$$\mathrm{dist}_\infty(\boldsymbol{\Phi}, \boldsymbol{X}^\star) = \mathcal{O}\left(\frac{\sqrt{\log n}}{p\sqrt{nq}}\right) \tag{22}$$

*holds with probability at least $1 - \mathcal{O}(1/n)$.*

*Proof.* Let $\boldsymbol{\Pi}^m = \mathrm{argmin}_{\boldsymbol{\Pi}\in\mathrm{SO}(d)} \|\boldsymbol{\Phi} - \boldsymbol{\Phi}^{(m)}\boldsymbol{\Pi}\|_F$. We can compute

$$\begin{aligned}
\|(\boldsymbol{W}\boldsymbol{\Phi})_m\|_F = \|\boldsymbol{W}_m^\top \boldsymbol{\Phi}\|_F &\leq \|\boldsymbol{W}_m^\top \boldsymbol{\Phi}^{(m)}\boldsymbol{\Pi}^m\|_F + \|\boldsymbol{W}_m^\top(\boldsymbol{\Phi} - \boldsymbol{\Phi}^{(m)}\boldsymbol{\Pi}^m)\|_F \\
&\leq \|\boldsymbol{W}_m^\top \boldsymbol{\Phi}^{(m)}\|_F + \|\boldsymbol{W}_m\|_2 \|\boldsymbol{\Phi} - \boldsymbol{\Phi}^{(m)}\boldsymbol{\Pi}^m\|_F.
\end{aligned} \tag{23}$$

The fact that $\boldsymbol{\Phi}^{(m)}$ is independent from $\boldsymbol{W}_m$ implies $\|\boldsymbol{W}_m^\top \boldsymbol{\Phi}^{(m)}\|_F = \mathcal{O}(\sqrt{nq\log n})$, so we have

$$\|(\boldsymbol{W}\boldsymbol{\Phi})_m\|_F = \mathcal{O}(\sqrt{nq\log n})\,(1 + \mathcal{O}(1)) = \mathcal{O}(\sqrt{nq\log n}). \tag{24}$$

Next, let $\boldsymbol{\Pi}^\star = \mathrm{argmin}_{\boldsymbol{\Pi}\in\mathrm{SO}(d)} \|\boldsymbol{\Phi} - \boldsymbol{X}^\star\boldsymbol{\Pi}\|_F$, then we have

$$(\boldsymbol{Y}\boldsymbol{\Phi})_m = pq\boldsymbol{X}_m^\star \boldsymbol{X}^{\star\top}\boldsymbol{\Phi} + (\boldsymbol{W}\boldsymbol{\Phi})_m = npq\boldsymbol{X}_m^\star\boldsymbol{\Pi}^\star + pq\boldsymbol{X}_m^\star\boldsymbol{X}^{\star\top}(\boldsymbol{\Phi} - \boldsymbol{X}^\star\boldsymbol{\Pi}^\star) + (\boldsymbol{W}\boldsymbol{\Phi})_m.$$

Since $\boldsymbol{\Phi}$ is the $d$-leading eigenvector of $\boldsymbol{Y}$, we have $\boldsymbol{Y}\boldsymbol{\Phi} = \boldsymbol{\Phi}\boldsymbol{\Sigma}$ with $\boldsymbol{\Sigma} \in \mathbb{R}^{d\times d}$ consisting of the $d$-leading eigenvalues of $\boldsymbol{Y}$. Applying the standard eigenvalue perturbation theory, e.g., [37, Theorem 4.11], we have

$$\|\boldsymbol{\Sigma} - npq\boldsymbol{I}\|_2 = \mathcal{O}(\|\boldsymbol{W}\|_2) = \mathcal{O}(\sqrt{nq\log n}).$$

Based on the assumption that $p^2 q = \Omega\left(\frac{\log n}{n}\right)$, we can show that $\boldsymbol{\Sigma} \geq \frac{3}{4}npq\boldsymbol{I}$. Note that $\boldsymbol{Y}\boldsymbol{\Phi} = \boldsymbol{\Phi}\boldsymbol{\Sigma}$ implies $\boldsymbol{\Phi}_m = (\boldsymbol{Y}\boldsymbol{\Phi})_m\boldsymbol{\Sigma}^{-1}$ for each $m \in [n]$, then we have

$$\|\boldsymbol{\Phi}_m\|_F \leq \frac{4}{3npq}\|(\boldsymbol{Y}\boldsymbol{\Phi})_m\|_F \leq \frac{4}{3npq}\|pq\boldsymbol{X}_m^\star\boldsymbol{X}^{\star\top}\boldsymbol{\Phi}_m\|_F + \frac{4}{3npq}\|(\boldsymbol{W}\boldsymbol{\Phi})_m\|_F \leq 2\sqrt{d},$$

where the last inequality holds because $\frac{4}{3npq}\|pq\boldsymbol{X}_m^\star\boldsymbol{X}^{\star\top}\boldsymbol{\Phi}_m\|_F = \frac{4}{3n}\|\boldsymbol{X}^{\star\top}\boldsymbol{\Phi}_m\|_F \leq \frac{4\sqrt{d}}{3}$ and (24). Therefore, for each $m \in [n]$,

$$\begin{aligned}
npq(\boldsymbol{\Phi}_m - \boldsymbol{X}_m^\star\boldsymbol{\Pi}^\star) &= \boldsymbol{\Phi}_m(npq\boldsymbol{I} - \boldsymbol{\Sigma}) + \boldsymbol{\Phi}_m\boldsymbol{\Sigma} - npq\boldsymbol{X}_m^\star\boldsymbol{\Pi}^\star \\
&= \boldsymbol{\Phi}_m(npq\boldsymbol{I} - \boldsymbol{\Sigma}) + (\boldsymbol{Y}\boldsymbol{\Phi})_m - npq\boldsymbol{X}_m^\star\boldsymbol{\Pi}^\star.
\end{aligned}$$

This further implies

$$\begin{aligned}
npq\|\boldsymbol{\Phi}_m - \boldsymbol{X}_m^\star\boldsymbol{\Pi}^\star\|_F &\leq \|\boldsymbol{\Sigma} - npq\boldsymbol{I}\|_2\|\boldsymbol{\Phi}_m\|_F + pq\|\boldsymbol{X}_m^\star\boldsymbol{X}^{\star\top}(\boldsymbol{\Phi} - \boldsymbol{X}^\star\boldsymbol{\Pi}^\star)\|_F + \|(\boldsymbol{W}\boldsymbol{\Phi})_m\|_F \\
&\leq 2\sqrt{d}\|\boldsymbol{W}\|_2 + \sqrt{n}pq\|\boldsymbol{\Phi} - \boldsymbol{X}^\star\boldsymbol{\Pi}^\star\|_F + \mathcal{O}\sqrt{nq(\log n)} \\
&= \mathcal{O}(\sqrt{nq\log n}),
\end{aligned}$$

where the last inequality holds because of Lemma 2 and Lemma 4. This completes the proof. $\square$

Finally, since $\boldsymbol{X}^0 = \mathcal{P}_{\mathrm{SO}(d)^n}(\boldsymbol{\Phi})$, according to Lemma 2 in [27], we have

$$\mathrm{dist}(\boldsymbol{X}^0, \boldsymbol{X}^\star) \le 2\,\mathrm{dist}(\boldsymbol{\Phi}, \boldsymbol{X}^\star) \quad \text{and} \quad \mathrm{dist}_\infty(\boldsymbol{X}^0, \boldsymbol{X}^\star) \le 2\,\mathrm{dist}_\infty(\boldsymbol{\Phi}, \boldsymbol{X}^\star),$$

which complete the proof of Theorem 2.

## B  Full Proof of Theorem 3

In this section, we provide the full proof of Lemma 1, Proposition 1, and Proposition 2, which finishes the proof of Theorem 3. To simplify the notation and the theoretical derivations, we assume without loss of generality that $\boldsymbol{X}_i^\star = \boldsymbol{I}$ for all $1 \le i \le n$ as one can separately rotate the space that each variable $\boldsymbol{X}_i$ lies in such that the corresponding ground-truth $\boldsymbol{X}_i^\star$ is rotated to identity [39, Lemma 4.1]. Consequently, we have $\boldsymbol{Y}_{ij} = \boldsymbol{I}$ for $(i, j) \in \mathcal{A}$.

### B.1  Proof of Lemma 1

For each $1 \le i \le n$, $|\mathcal{E}_i| = \sum_{1 \le j \le n} \mathbf{1}_{\mathcal{E}_i}(j)$, where $\mathbf{1}_{\mathcal{E}_i}(\cdot)$ denotes the indicator function w.r.t $\mathcal{E}_i$. Based on our model, $\sum_{1 \le j \le n} \mathbf{1}_{\mathcal{E}_i}(j)$ follows the binomial distribution $B(n, q)$. According to the Bernstein inequality [38], for any constant $\epsilon \in (0, 1)$, we have

$$\Pr\left( \left| \sum_{1 \le j \le n} \mathbf{1}_{\mathcal{E}_i}(j) - nq \right| \ge \epsilon nq \right) \le 2\exp\left( -\frac{\frac{1}{2}\epsilon^2 n^2 q^2}{\sum_{1 \le j \le n} E\{\mathbf{1}_{\mathcal{E}_i}^2(j)\} + \epsilon nq/3} \right)$$

$$= 2\exp\left( -\frac{\frac{1}{2}\epsilon^2 n^2 q^2}{nq + \epsilon nq/3} \right) \le 2\exp\left( -\frac{3}{8}\epsilon^2 nq \right).$$

The last inequality holds because of $\epsilon < 1$. Therefore,

$$\Pr\left( \bigcup_{1 \le i \le n} \{|\mathcal{E}_i| - nq| \ge \epsilon nq\} \right) \le 2n\exp\left( -\frac{3}{8}\epsilon^2 nq \right) \le \frac{2}{n^2}, \tag{25}$$

where the last inequality holds because we assume that $\epsilon \ge \frac{\sqrt{8\log n}}{\sqrt{npq}}$. Similarly, we have

$$\Pr\left( \bigcup_{1 \le i \le n} \{|\mathcal{A}_i| - npq| \ge \epsilon npq\} \right) \le 2n\exp\left( -\frac{3}{8}\epsilon^2 npq \right) \le \frac{2}{n^2}, \tag{26}$$

$$\Pr\left( \bigcup_{1 \le i,j \le n} \{|\mathcal{E}_i \cap \mathcal{A}_j| - npq^2| \ge \epsilon npq^2\} \right) \le 2n^2\exp\left( -\frac{3}{8}\epsilon^2 npq^2 \right) \le \frac{2}{n}, \tag{27}$$

and

$$\Pr\left( \bigcup_{1 \le i,j \le n} \{|\mathcal{A}_{ij}| - np^2q^2| \ge \epsilon np^2q^2\} \right) \le 2n^2\exp\left( -\frac{3}{8}\epsilon^2 np^2q^2 \right) \le \frac{2}{n}. \tag{28}$$

Hence, we complete the proof of Lemma 1 once $n \ge 4$.

### B.2  Proof of Proposition 1

We can first compute

$$\sum_{1 \le i,j \le n} \|\boldsymbol{X}_i - \boldsymbol{X}_j\|_F \le \sum_{1 \le i,j \le n} \frac{1}{|\mathcal{A}_{ij}|} \sum_{k \in \mathcal{A}_{ij}} (\|\boldsymbol{X}_i - \boldsymbol{X}_k\|_F + \|\boldsymbol{X}_j - \boldsymbol{X}_k\|_F)$$

$$\le \frac{1}{(1-\epsilon)np^2q^2} \sum_{1 \le i,j \le n} \sum_{k \in \mathcal{A}_{ij}} (\|\boldsymbol{X}_i - \boldsymbol{X}_k\|_F + \|\boldsymbol{X}_j - \boldsymbol{X}_k\|_F). \tag{29}$$

Here, the first inequality comes from the triangle inequality, while the second one follows from Lemma 1. Now, invoking Lemma 1, which tells $|\mathcal{A}_k| \le (1 + \epsilon)npq$, gives

$$\sum_{1 \le i,j \le n} \sum_{k \in \mathcal{A}_{ij}} \|\boldsymbol{X}_i - \boldsymbol{X}_k\|_F = \sum_{1 \le i \le n} \sum_{k \in \mathcal{A}_i} \sum_{j \in \mathcal{A}_k} \|\boldsymbol{X}_i - \boldsymbol{X}_k\|_F$$

$$\le (1 + \epsilon)npq \sum_{1 \le i \le n} \sum_{k \in \mathcal{A}_i} \|\boldsymbol{X}_i - \boldsymbol{X}_k\|_F$$

$$= (1+\epsilon)npq \sum_{(i,k)\in\mathcal{A}} \|\boldsymbol{X}_i - \boldsymbol{X}_k\|_F.$$

By symmetry, we conclude that

$$\sum_{1\leq i,j\leq n}\sum_{k\in\mathcal{A}_{ij}} (\|\boldsymbol{X}_i - \boldsymbol{X}_k\|_F + \|\boldsymbol{X}_j - \boldsymbol{X}_k\|_F) \leq 2(1+\epsilon)npq \sum_{(i,j)\in\mathcal{A}} \|\boldsymbol{X}_i - \boldsymbol{X}_j\|_F. \tag{30}$$

Furthermore, we claim the following bound for any $\boldsymbol{X} \in \mathrm{SO}(d)^n$:

$$\sum_{1\leq i,j\leq n} \|\boldsymbol{X}_i - \boldsymbol{X}_j\|_F \geq \frac{n}{2}\,\mathrm{dist}_1(\boldsymbol{X}, \boldsymbol{X}^\star). \tag{31}$$

Combining (29), (30), (31), and the fact $g(\boldsymbol{X}) = \sum_{(i,j)\in\mathcal{A}} \|\boldsymbol{X}_i - \boldsymbol{X}_j\|_F$ establishes Proposition 1.

Hence, it remains to show (31). First of all, according to the triangle inequality, we have

$$\sum_{1\leq i,j\leq n} \|\boldsymbol{X}_i - \boldsymbol{X}_j\|_F \geq \sum_{1\leq i\leq n} \left\| n\boldsymbol{X}_i - \sum_{1\leq j\leq n} \boldsymbol{X}_j \right\|_F = n \sum_{1\leq i\leq n} \|\boldsymbol{X}_i - \overline{\boldsymbol{X}}\|_F, \tag{32}$$

where $\overline{\boldsymbol{X}} = \frac{1}{n}\sum_{1\leq j\leq n} \boldsymbol{X}_j$ can be taken as a diagonal matrix (since the Frobenius norm is invariant up to a global rotation) with its first $(d-1)$ diagonal entries being positive. Finally, applying the following lemma to (32) provides (31).

**Lemma 7.** *For any $\boldsymbol{A} \in \mathrm{SO}(d)$ and $\boldsymbol{B} = Diag(b_1,\ldots,b_d)$ satisfying $b_1,\ldots,b_{d-1} \in [0,1]$ and $b_d \in [-1,1]$, we have*

$$\|\boldsymbol{A} - \boldsymbol{B}\|_F \geq \frac{1}{2}\|\boldsymbol{A} - \boldsymbol{I}\|_F.$$

*Proof.* It is equivalent to show that $\|\boldsymbol{A} - \boldsymbol{B}\|_F^2 \geq \frac{1}{4}\|\boldsymbol{A} - \boldsymbol{I}\|_F^2$. Since $\boldsymbol{A} \in \mathrm{SO}(d)$, we simplify as

$$d + \langle \boldsymbol{A}, \boldsymbol{I} - 4\boldsymbol{B} \rangle + 2\|\boldsymbol{B}\|_F^2 = \sum_{1\leq i\leq d} 1 + \boldsymbol{A}_{ii}(1 - 4b_i) + 2b_i^2 \geq 0. \tag{33}$$

To prove (33) holds for all $\boldsymbol{A} \in \mathrm{SO}(d)$, we choose $\bar{\boldsymbol{A}} = \mathrm{argmin}_{\boldsymbol{A}\in\mathrm{SO}(d)}\langle \boldsymbol{A}, \boldsymbol{I} - 4\boldsymbol{B}\rangle = \mathcal{P}_{\mathrm{SO}(d)}(4\boldsymbol{B} - \boldsymbol{I})$. It is easy to see that $\bar{\boldsymbol{A}}$ is also a diagonal matrix. For any $1 \leq i \leq d-1$, we have

$$1 + \boldsymbol{A}_{ii}(1 - 4b_i) + 2b_i^2 = \begin{cases} 2(b_i - 1)^2, & \boldsymbol{A}_{ii} = 1; \\ 2b_i^2 + 4b_i, & \boldsymbol{A}_{ii} = -1. \end{cases}$$

So it is always nonnegative since $b_i \geq 0$ for any $1 \leq i \leq d-1$. For the last summation in (33), on the one hand, if $\bar{\boldsymbol{A}}_{dd} = 1$, then $1 + \boldsymbol{A}_{dd}(1 - 4b_d) + 2b_d^2 = 2(b_d - 1)^2 \geq 0$. On the other hand, if $\bar{\boldsymbol{A}}_{dd} = -1$, then there exist another $1 \leq k \leq d-1$ such that $\bar{\boldsymbol{A}}_{kk} = -1$. So we have $(1 + \boldsymbol{A}_{kk}(1 - 4b_k) + 2b_k^2) + (1 + \boldsymbol{A}_{dd}(1 - 4b_d) + 2b_d^2) = 2b_k^2 + 2b_d^2 + 4(b_k + b_d) \geq 0$. The last inequality holds because $|b_d| \leq b_k$. We complete the proof. $\square$

## B.3 Proof of Proposition 2

Based on (11) and our simplification that $\boldsymbol{X}_i^\star = \boldsymbol{I}$, $1 \leq i \leq n$, we have

$$h(\boldsymbol{X}) - h(\boldsymbol{X}^\star) \geq \sum_{(i,j)\in\mathcal{A}^c} \left\langle \frac{\boldsymbol{I} - \boldsymbol{O}_{ij}}{\|\boldsymbol{I} - \boldsymbol{O}_{ij}\|_F}, \boldsymbol{X}_i^\top \boldsymbol{X}_j - \boldsymbol{I} \right\rangle. \tag{34}$$

Let $\boldsymbol{A} = \mathbb{E}\left\{\frac{\boldsymbol{I} - \boldsymbol{O}_{ij}}{\|\boldsymbol{I} - \boldsymbol{O}_{ij}\|_F}\right\}$, $\boldsymbol{Z}_{ij} = \frac{\boldsymbol{I} - \boldsymbol{O}_{ij}}{\|\boldsymbol{I} - \boldsymbol{O}_{ij}\|_F} \cdot \boldsymbol{1}_{\mathcal{A}^c}(ij) - (1-p)q\boldsymbol{A}$, and $\boldsymbol{Z} \in \mathbb{R}^{nd\times nd}$ collects each $\boldsymbol{Z}_{ij}$ in its $(i,j)$-th block. We can compute

$$\left| \sum_{(i,j)\in\mathcal{A}^c} \left\langle \frac{\boldsymbol{I} - \boldsymbol{O}_{ij}}{\|\boldsymbol{I} - \boldsymbol{O}_{ij}\|_F}, \boldsymbol{X}_i^\top \boldsymbol{X}_j - \boldsymbol{I} \right\rangle \right| = \left| \sum_{1\leq i,j\leq n} \langle \boldsymbol{Z}_{ij} + (1-p)q\boldsymbol{A}, \boldsymbol{X}_i^\top \boldsymbol{X}_j - \boldsymbol{I} \rangle \right|$$

$$\leq \left| \sum_{1\leq i,j\leq n} \langle (1-p)q\boldsymbol{A}, \boldsymbol{X}_i^\top \boldsymbol{X}_j - \boldsymbol{I} \rangle \right| \tag{35}$$

$$+ \left| \sum_{1 \le i,j \le n} \left\langle \boldsymbol{Z}_{ij}, (\boldsymbol{X}_i - \boldsymbol{I})^\top (\boldsymbol{X}_j - \boldsymbol{I}) + (\boldsymbol{X}_i - \boldsymbol{I})^\top + (\boldsymbol{X}_j - \boldsymbol{I}) \right\rangle \right|.$$

To bound the first term in (35), we can proceed as

$$
\begin{aligned}
\left| \sum_{1 \le i,j \le n} \left\langle (1-p)q\boldsymbol{A}, \boldsymbol{X}_i^\top \boldsymbol{X}_j - \boldsymbol{I} \right\rangle \right| &= \left| \left\langle (1-p)q\boldsymbol{A}, \left( \sum_{i=1}^n \boldsymbol{X}_i \right)^\top \left( \sum_{i=1}^n \boldsymbol{X}_i \right) - n^2 \boldsymbol{I} \right\rangle \right| \\
&= \left| \left\langle (1-p)q\boldsymbol{A}, \left( \sum_{i=1}^n \boldsymbol{X}_i + n\boldsymbol{I} \right)^\top \left( \sum_{i=1}^n \boldsymbol{X}_i - n\boldsymbol{I} \right) \right\rangle \right| \\
&\le \left\| (1-p)q\boldsymbol{A} \left( \sum_{i=1}^n \boldsymbol{X}_i + n\boldsymbol{I} \right) \right\|_F \left\| \sum_{i=1}^n \boldsymbol{X}_i - n\boldsymbol{I} \right\|_F \\
&\le \| (1-p)q\boldsymbol{A} \|_F \left\| \sum_{i=1}^n \boldsymbol{X}_i + n\boldsymbol{I} \right\|_2 \left\| \sum_{i=1}^n \boldsymbol{X}_i - n\boldsymbol{I} \right\|_F \\
&\le 2(1-p)qn \|\boldsymbol{A}\|_F \left\| \sum_{i=1}^n \boldsymbol{X}_i - n\boldsymbol{I} \right\|_F.
\end{aligned}
$$
(36)

Here, the second equality is true since $\sum_{i=1}^n \boldsymbol{X}_i$ can be taken as a diagonal matrix. According to [39, Lemma A.1], we know that $\|\boldsymbol{A}\|_F = \left\| \mathbb{E} \left\{ \frac{\boldsymbol{I} - \boldsymbol{O}_{ij}}{\|\boldsymbol{I} - \boldsymbol{O}_{ij}\|_F} \right\} \right\|_F \le \frac{1}{\sqrt{2}}$ for all $d \ge 2$. On the other hand, we know that

$$\text{dist}(\boldsymbol{X}, \boldsymbol{X}^\star)^2 = \sum_{i=1}^n \|\boldsymbol{X}_i - \boldsymbol{I}\|_F^2 = 2\,\text{trace} \left( n\boldsymbol{I} - \sum_{i=1}^n \boldsymbol{X}_i \right) \ge 2 \left\| \sum_{i=1}^n \boldsymbol{X}_i - n\boldsymbol{I} \right\|_F, \qquad (37)$$

where the last inequality holds because $n\boldsymbol{I} - \sum_{i=1}^n \boldsymbol{X}_i$ is a nonnegative diagonal matrix. Therefore, we obtain

$$
\begin{aligned}
\left| \sum_{1 \le i,j \le n} \left\langle (1-p)q\boldsymbol{A}, \boldsymbol{X}_i^\top \boldsymbol{X}_j - \boldsymbol{I} \right\rangle \right| &\le \frac{(1-p)qn}{\sqrt{2}} \,\text{dist}(\boldsymbol{X}, \boldsymbol{X}^\star)^2 \\
&\le \frac{(1-p)qn}{\sqrt{2}} \max_i \|\boldsymbol{X}_i - \boldsymbol{I}\|_F \sum_{i=1}^n \|\boldsymbol{X}_i - \boldsymbol{I}\|_F.
\end{aligned}
$$
(38)

To further bound the second term in (35), we can proceed as

$$
\begin{aligned}
&\left| \sum_{1 \le i,j \le n} \left\langle \boldsymbol{Z}_{ij}, (\boldsymbol{X}_i - \boldsymbol{I})^\top (\boldsymbol{X}_j - \boldsymbol{I}) + (\boldsymbol{X}_i - \boldsymbol{I})^\top + (\boldsymbol{X}_j - \boldsymbol{I}) \right\rangle \right| \\
&\le (\boldsymbol{X} - \boldsymbol{I}^n)\boldsymbol{Z}(\boldsymbol{X} - \boldsymbol{I}^n)^\top + 2 \left| \sum_{1 \le i \le n} \left\langle \sum_{1 \le j \le n} \boldsymbol{Z}_{ij}, \boldsymbol{X}_i - \boldsymbol{I} \right\rangle \right| \\
&\le \|\boldsymbol{Z}\|_{op} \|\boldsymbol{X} - \boldsymbol{I}^n\|_F^2 + 2 \sum_{1 \le i \le n} \left\| \sum_{1 \le j \le n} \boldsymbol{Z}_{ij} \right\|_F \|\boldsymbol{X}_i - \boldsymbol{I}\|_F \\
&\le \left( 2\sqrt{d} \|\boldsymbol{Z}\|_{op} + 2 \max_{1 \le i \le n} \left\| \sum_{1 \le j \le n} \boldsymbol{Z}_{ij} \right\|_F \right) \sum_{1 \le i \le n} \|\boldsymbol{X}_i - \boldsymbol{I}\|_F,
\end{aligned}
$$
(39)

where $\boldsymbol{I}^n \in \mathbb{R}^{nd \times d}$ collects $n$ identity matrix together and the last inequality holds because $\|\boldsymbol{X} - \boldsymbol{I}^n\|_F^2 = \sum_{1 \le i \le n} \|\boldsymbol{X}_i - \boldsymbol{I}\|_F^2 \le \sum_{1 \le i \le n} (\|\boldsymbol{X}_i\|_F + \|\boldsymbol{I}\|_F) \|\boldsymbol{X}_i - \boldsymbol{I}\|_F = 2\sqrt{d} \sum_{1 \le i \le n} \|\boldsymbol{X}_i - \boldsymbol{I}\|_F$.

Using the randomness of $\boldsymbol{O}_{ij}$, we claim that with probability at least $1 - 4d/n$, we have

$$\|\boldsymbol{Z}\|_{\mathrm{op}} \leq \sqrt{8n(1-p)q\log n}, \quad \text{and} \quad \max_{1 \leq i \leq n} \left\| \sum_{1 \leq j \leq n} \boldsymbol{Z}_{ij} \right\|_F \leq \sqrt{8n(1-p)q\log n}. \tag{40}$$

Combining the above bounds gives

$$h(\boldsymbol{X}) - h(\boldsymbol{X}^\star)$$
$$\geq -\left( 2(\sqrt{d}+1)\sqrt{8n(1-p)q\log n} + \frac{(1-p)qn}{\sqrt{2}} \max_i \|\boldsymbol{X}_i - \boldsymbol{I}\|_F \right) \sum_{1 \leq i \leq n} \|\boldsymbol{X}_i - I\|_F \tag{41}$$
$$\geq -\frac{npq}{16} \cdot \mathrm{dist}_1(\boldsymbol{X}, \boldsymbol{X}^\star),$$

where the last inequality holds because we assume $p^2 q^2 = \Omega\left(\frac{\log n}{n}\right)$, $\max_i \|\boldsymbol{X}_i - \boldsymbol{I}\|_F = \mathrm{dist}_\infty(\boldsymbol{X}, \boldsymbol{X}^\star) = \mathcal{O}(p)$, and $\sum_{1 \leq i \leq n} \|\boldsymbol{X}_i - I\|_F = \mathrm{dist}_1(\boldsymbol{X}, \boldsymbol{X}^\star)$.

Finally, it remains to show that the two inequalities in (40) holds with probability at least $1 - 4d/n$. It is quick to verify that $\mathbb{E}(\boldsymbol{Z}_{ij}) = \boldsymbol{0}, \|\boldsymbol{Z}_{ij}\|_{op} \leq 1 + (1-p)q\|\boldsymbol{A}\|_F \leq 2$, and

$$\mathbb{E}(\boldsymbol{Z}^2) = \mathrm{BlkDiag}\left( \sum_j \mathbb{E}(\boldsymbol{Z}_{1j}\boldsymbol{Z}_{1j}^\top), \ldots, \sum_j \mathbb{E}(\boldsymbol{Z}_{nj}\boldsymbol{Z}_{nj}^\top) \right)$$
$$= \sum_j \mathbb{E}(\boldsymbol{Z}_{1j}\boldsymbol{Z}_{1j}^\top) \otimes \boldsymbol{I}_n$$
$$= n(1-p)q\mathbb{E}\left( \frac{(\boldsymbol{I}-\boldsymbol{O}_{ij})(\boldsymbol{I}-\boldsymbol{O}_{ij})^\top}{\|\boldsymbol{I}-\boldsymbol{O}_{ij}\|_F^2} - (1-p)^2 q^2 \boldsymbol{A}\boldsymbol{A}^\top \right) \otimes \boldsymbol{I}_n,$$

where $\mathrm{BlkDiag}(\cdot)$ means the block diagonal matrix, $\otimes$ means the Kronecker product and $\boldsymbol{I}_n$ denotes the $n$-by-$n$ identity matrix. Thus, we have $\|\mathbb{E}(\boldsymbol{Z}^2)\|_{op} \leq n(1-p)q$. According to the Matrix Bernstein inequality [38], we have

$$\Pr\left( \|\boldsymbol{Z}\|_{\mathrm{op}} \leq \sqrt{8n(1-p)q\log n} \right) \geq 1 - 2nd \exp\left( \frac{-4n(1-p)q\log n}{n(1-p)q + 2\sqrt{8n(1-p)q\log n}/3} \right)$$
$$\geq 1 - \frac{2d}{n}. \tag{42}$$

Here, the last inequality holds because we assume $2\sqrt{8n(1-p)q\log n}/3 \leq n(1-p)q$, which is true as long as $p = \Omega(\log n/n)$.

For any fixed $1 \leq i \leq n$, a similar argument based on Matrix Bernstein inequality [38] shows that

$$\Pr\left( \|\sum_j \boldsymbol{Z}_{ij}\|_F \leq \sqrt{8n(1-p)q\log n} \right) \geq 1 - 2d \exp\left( \frac{-4n(1-p)q\log n}{n(1-p)q + \sqrt{8n(1-p)q\log n}/3} \right)$$
$$\geq 1 - \frac{2d}{n^2}, \tag{43}$$

which implies $\Pr\left( \max_i \|\sum_j \boldsymbol{Z}_{ij}\|_F \leq \sqrt{8n(1-p)q\log n} \right) \geq 1 - 2d/n$.

## C  Full Proof of Theorem 4

### C.1  Proof of Corollary 1

Combining the convexity of $f$ and Theorem 3, we have

$$-\frac{npq}{8} \sum_{1 \leq i \leq n} \|\boldsymbol{X}_i - \boldsymbol{X}_i^\star\|_F \geq f(\boldsymbol{X}^\star) - f(\boldsymbol{X}) \geq \left\langle \widetilde{\nabla} f(\boldsymbol{X}), \boldsymbol{X}^\star - \boldsymbol{X} \right\rangle, \quad \forall \widetilde{\nabla} f(\boldsymbol{X}) \in \partial f(\boldsymbol{X}). \tag{44}$$

for all $\boldsymbol{X} \in \mathrm{SO}(d)^n$ satisfying $\mathrm{dist}_\infty(\boldsymbol{X}, \boldsymbol{X}^\star) = \mathcal{O}(p)$. For any $\widetilde{\nabla}_{\mathcal{R}}^\perp f(\boldsymbol{X}) \in \partial_{\mathcal{R}}^\perp f(\boldsymbol{X})$, we can further compute

$$
\begin{aligned}
\left\langle \widetilde{\nabla}_{\mathcal{R}}^\perp f(\boldsymbol{X}), \boldsymbol{X} - \boldsymbol{X}^\star \right\rangle &= \sum_{1 \leq i \leq n} \left\langle \widetilde{\nabla}_{\mathcal{R}}^\perp f(\boldsymbol{X}_i), \mathcal{P}_{\mathrm{T}_{\boldsymbol{X}_i}^\perp}(\boldsymbol{X}_i - \boldsymbol{X}_i^\star) \right\rangle \\
&\leq \sum_{1 \leq i \leq n} \|\widetilde{\nabla}_{\mathcal{R}}^\perp f(\boldsymbol{X}_i)\|_F \cdot \|\mathcal{P}_{\mathrm{T}_{\boldsymbol{X}_i}^\perp}(\boldsymbol{X}_i - \boldsymbol{X}_i^\star)\|_F.
\end{aligned}
$$

On the one hand, notice that

$$
\begin{aligned}
\mathcal{P}_{\mathrm{T}_{\boldsymbol{X}_i}^\perp}(\boldsymbol{X}_i - \boldsymbol{X}_i^\star) &= \boldsymbol{X}_i - \boldsymbol{X}_i^\star - \mathcal{P}_{\mathrm{T}_{\boldsymbol{X}_i}}(\boldsymbol{X}_i - \boldsymbol{X}_i^\star) \\
&= \boldsymbol{X}_i \left( \boldsymbol{X}_i^\top (\boldsymbol{X}_i - \boldsymbol{X}_i^\star) + (\boldsymbol{X}_i - \boldsymbol{X}_i^\star)^\top \boldsymbol{X}_i \right) / 2 \\
&= \boldsymbol{X}_i (\boldsymbol{X}_i - \boldsymbol{X}_i^\star)^\top (\boldsymbol{X}_i - \boldsymbol{X}_i^\star) / 2,
\end{aligned}
$$

which implies

$$
\|\mathcal{P}_{\mathrm{T}_{\bar{\boldsymbol{X}}}^\perp}(\boldsymbol{X}_i - \boldsymbol{X}_i^\star)\|_F = \frac{1}{2} \|\boldsymbol{X}_i (\boldsymbol{X}_i - \boldsymbol{X}_i^\star)^\top (\boldsymbol{X}_i - \boldsymbol{X}_i^\star)\|_F \leq \frac{1}{2} \|\boldsymbol{X}_i - \boldsymbol{X}_i^\star\|_F^2.
$$

On the other hand, according to Lemma 1, with probability at least $1 - \mathcal{O}(1/n)$, for any $i \in [n]$,

$$
\|\widetilde{\nabla}_{\mathcal{R}}^\perp f(\boldsymbol{X}_i)\|_F \leq \|\widetilde{\nabla} f(\boldsymbol{X}_i)\|_F = \left\| \sum_{j \in \mathcal{E}_i} \widetilde{\nabla} f_{i,j}(\boldsymbol{X}_i) \right\|_F \leq \sum_{j \in \mathcal{E}_i} \left\| \widetilde{\nabla} f_{i,j}(\boldsymbol{X}_i) \right\|_F \leq |\mathcal{E}_i| \leq 2nq,
$$

where $\widetilde{\nabla} f_{i,j}(\boldsymbol{X}_i) \in \partial f_{i,j}(\boldsymbol{X}_i)$ satisfies $\|\widetilde{\nabla} f_{i,j}(\boldsymbol{X}_i)\|_F \leq 1$. Hence, we have

$$
\left\langle \widetilde{\nabla}_{\mathcal{R}}^\perp f(\boldsymbol{X}), \boldsymbol{X} - \boldsymbol{X}^\star \right\rangle \leq nq \sum_{1 \leq i \leq n} \|\boldsymbol{X}_i - \boldsymbol{X}_i^\star\|_F^2 \leq \frac{npq}{16} \sum_i \|\boldsymbol{X}_i - \boldsymbol{X}_i^\star\|_F
$$

for any $\boldsymbol{X}$ such that $\mathrm{dist}_\infty(\boldsymbol{X}, \boldsymbol{X}^\star) \leq \frac{p}{16}$. Invoking the above bounds into (44) yields the desired result

$$
\left\langle \widetilde{\nabla}_{\mathcal{R}} f(\boldsymbol{X}), \boldsymbol{X} - \boldsymbol{X}^\star \right\rangle = \left\langle \widetilde{\nabla} f(\boldsymbol{X}) - \widetilde{\nabla}_{\mathcal{R}}^\perp f(\boldsymbol{X}), \boldsymbol{X} - \boldsymbol{X}^\star \right\rangle \geq \frac{npq}{16} \sum_{1 \leq i \leq n} \|\boldsymbol{X}_i - \boldsymbol{X}_i^\star\|_F.
$$

### C.2 Proof of Contraction

Let us first present some preliminary results, which will be used in our later derivations. By noticing that $\widetilde{\nabla} f(\boldsymbol{X}_i) = \sum_{j \in \mathcal{E}_i} \widetilde{\nabla} f_{i,j}(\boldsymbol{X}_i)$ where $\widetilde{\nabla} f_{i,j}(\boldsymbol{X}_i) \in \partial f_{i,j}(\boldsymbol{X}_i)$, we define

$$
\widetilde{\nabla} g(\boldsymbol{X}_i) = \sum_{j \in \mathcal{A}_i} \widetilde{\nabla} f_{i,j}(\boldsymbol{X}_i), \quad \widetilde{\nabla} h(\boldsymbol{X}_i) = \sum_{j \in \mathcal{E}_i \setminus \mathcal{A}_i} \widetilde{\nabla} f_{i,j}(\boldsymbol{X}_i).
$$

Recall that $\widetilde{\nabla}_{\mathcal{R}} g(\boldsymbol{X}_i) = \mathcal{P}_{\mathrm{T}_{\boldsymbol{X}_i}}(\widetilde{\nabla} g(\boldsymbol{X}_i))$ and $\widetilde{\nabla}_{\mathcal{R}} h(\boldsymbol{X}_i) = \mathcal{P}_{\mathrm{T}_{\boldsymbol{X}_i}}(\widetilde{\nabla} h(\boldsymbol{X}_i))$. Similarly, we have $\widetilde{\nabla} f(\boldsymbol{X}_i) = \widetilde{\nabla} g(\boldsymbol{X}_i) + \widetilde{\nabla} h(\boldsymbol{X}_i)$ and $\widetilde{\nabla}_{\mathcal{R}} f(\boldsymbol{X}_i) = \widetilde{\nabla}_{\mathcal{R}} g(\boldsymbol{X}_i) + \widetilde{\nabla}_{\mathcal{R}} h(\boldsymbol{X}_i)$. Furthermore, the QR decomposition-based retraction satisfies the second-order boundedness property, i.e., there exists some $M \geq 1$ such that

$$
\begin{aligned}
\|\boldsymbol{X}_i^{k+1} - \boldsymbol{X}_i^\star\|_F &= \|\mathrm{Retr}_{\boldsymbol{X}_i^k}\left(-\mu_k \widetilde{\nabla}_{\mathcal{R}} f(\boldsymbol{X}_i^k)\right) - \boldsymbol{X}_i^\star\|_F \\
&\leq \|\boldsymbol{X}_i^k - \mu_k \widetilde{\nabla}_{\mathcal{R}} f(\boldsymbol{X}_i^k) - \boldsymbol{X}_i^\star\|_F + M \cdot \mu_k^2 \|\widetilde{\nabla}_{\mathcal{R}} f(\boldsymbol{X}_i^k)\|_F^2.
\end{aligned}
\tag{45}
$$

Recall that $\widetilde{\nabla}_{\mathcal{R}} f(\boldsymbol{X}_i^k) = \widetilde{\nabla}_{\mathcal{R}} g(\boldsymbol{X}_i^k) + \widetilde{\nabla}_{\mathcal{R}} h(\boldsymbol{X}_i^k)$, we have

$$
\begin{aligned}
\|\widetilde{\nabla}_{\mathcal{R}} f(\boldsymbol{X}_i^k)\|_F &\leq \|\widetilde{\nabla}_{\mathcal{R}} g(\boldsymbol{X}_i^k)\|_F + \|\widetilde{\nabla}_{\mathcal{R}} h(\boldsymbol{X}_i^k)\|_F \\
&\leq 5\sqrt{2\delta_0} nq + (1 + \epsilon) npq \leq (1 + 2\epsilon) npq,
\end{aligned}
\tag{46}
$$

where the second inequality comes from

$$
\|\widetilde{\nabla}_{\mathcal{R}} g(\boldsymbol{X}_i^k)\|_F = \left\| \sum_{j \in \mathcal{A}_i} \widetilde{\nabla}_{\mathcal{R}} f_{i,j}(\boldsymbol{X}_i) \right\|_F \leq \sum_{j \in \mathcal{A}_i} \left\| \widetilde{\nabla}_{\mathcal{R}} f_{i,j}(\boldsymbol{X}_i) \right\|_F \leq |\mathcal{A}_i| \leq (1 + \epsilon) npq,
$$

and Lemma 10 and the last inequality is due to the choice $\delta_0 \leq \epsilon^2 p^2/50$ (i.e., $\delta_0 = \mathcal{O}(p^2)$). Thus, by choosing $\mu_k = \mathcal{O}(\frac{\delta_k}{n})$, and $\delta_0 = \mathcal{O}(p^2)$, the second-order term $M \cdot \mu_k^2 \|\widetilde{\nabla}_{\mathcal{R}} f(\boldsymbol{X}_i^k)\|_F^2 = \mathcal{O}(p^6 q^2)$, which is a very high-order error. In the following analysis, we will ignore this term to simplify our derivations.

Using the above preliminaries and Corollary 1, we are ready to establish two key lemmas, which show that if $\boldsymbol{X}^k \in \mathcal{N}_F^k \cap \mathcal{N}_\infty^k$, then $\boldsymbol{X}^{k+1} \in \mathcal{N}_F^{k+1}$ (Lemma 8) and $\boldsymbol{X}^{k+1} \in \mathcal{N}_\infty^{k+1}$ (Lemma 9), respectively. This completes the proof of Theorem 4.

**Lemma 8.** *With high probability, suppose that $\boldsymbol{X}^k \in \mathcal{N}_F^k \cap \mathcal{N}_\infty^k$, $\mu_k = \mathcal{O}(\frac{\delta_k}{n})$, and*

$$\delta_0 = \mathcal{O}(p^2), \quad and \quad \xi_0 = \Theta(\sqrt{npq}\delta_0), \tag{47}$$

*then $\boldsymbol{X}^{k+1} \in \mathcal{N}_F^{k+1}$.*

*Proof.* By ignoring the high-order error term in (45), in order to bound $\|\boldsymbol{X}^{k+1} - \boldsymbol{X}^\star\|_F^2$ we can first compute

$$\|\boldsymbol{X}^{k+1} - \boldsymbol{X}^\star\|_F^2 = \sum_{1 \leq i \leq n} \|\boldsymbol{X}_i^{k+1} - \boldsymbol{X}_i^\star\|_F^2 \leq \sum_{1 \leq i \leq n} \|\boldsymbol{X}_i^k - \mu_k \widetilde{\nabla}_{\mathcal{R}} f(\boldsymbol{X}_i^k) - \boldsymbol{X}_i^\star\|_F^2$$

$$= \|\boldsymbol{X}^k - \boldsymbol{X}^\star\|_F^2 - 2\mu_k \left\langle \widetilde{\nabla}_{\mathcal{R}} f(\boldsymbol{X}^k), \boldsymbol{X}^k - \boldsymbol{X}^\star \right\rangle + \mu_k^2 \|\widetilde{\nabla}_{\mathcal{R}} f(\boldsymbol{X}^k)\|_F^2.$$

Then, according to Corollary 1, we have

$$\left\langle \widetilde{\nabla}_{\mathcal{R}} f(\boldsymbol{X}^k), \boldsymbol{X}^k - \boldsymbol{X}^\star \right\rangle \geq \frac{npq}{16} \sum_{1 \leq i \leq n} \|\boldsymbol{X}_i^k - \boldsymbol{X}_i^\star\|_F \geq \frac{npq}{16\delta_k} \sum_{1 \leq i \leq n} \|\boldsymbol{X}_i^k - \boldsymbol{X}_i^\star\|_F^2$$

$$= \frac{npq}{16\delta_k} \|\boldsymbol{X}^k - \boldsymbol{X}^\star\|_F^2,$$

where the second inequality holds because $\|\boldsymbol{X}_i^k - \boldsymbol{X}_i^\star\|_F \leq \delta_k$ (i.e., $\boldsymbol{X}^k \in \mathcal{N}_\infty^k$). Combining the above two inequalities gives

$$\|\boldsymbol{X}^{k+1} - \boldsymbol{X}^\star\|_F^2 \leq \left(1 - \mu_k \cdot \frac{npq}{8\delta_k}\right) \|\boldsymbol{X}^k - \boldsymbol{X}^\star\|_F^2 + \mu_k^2 \|\widetilde{\nabla}_{\mathcal{R}} f(\boldsymbol{X}^k)\|_F^2$$

$$\leq \left(1 - \frac{pq}{8}\right) \|\boldsymbol{X}^k - \boldsymbol{X}^\star\|_F^2 + (1 + 2\epsilon)^2 n^3 p^2 q^2 \mu_k^2,$$

where the last inequality is due to (46). Since $\mu_k = \mathcal{O}(\frac{\delta_k}{n})$ and $\xi_0 = \Theta(\sqrt{npq}\delta_0)$ (i.e., $\xi_k = \Theta(\sqrt{npq}\delta_k)$), we have $n^3 p^2 q^2 \mu_k^2 = \mathcal{O}(pq\xi_k^2)$, which implies

$$\|\boldsymbol{X}^{k+1} - \boldsymbol{X}^\star\|_F^2 \leq \left(1 - \frac{pq}{8}\right)\xi_k^2 + \frac{pq}{16}\xi_k^2 = \left(1 - \frac{pq}{16}\right)\xi_k^2 = \xi_{k+1}^2.$$

This completes the proof. $\qquad\square$

**Lemma 9.** *With high probability, suppose that $\boldsymbol{X}^k \in \mathcal{N}_F^k \cap \mathcal{N}_\infty^k$, $\mu_k = \mathcal{O}(\frac{\delta_k}{n})$, and*

$$\delta_0 = \mathcal{O}(p^2) \quad and \quad \xi_0 = \mathcal{O}(\sqrt{npq}\delta_0), \tag{48}$$

*then $\boldsymbol{X}^{k+1} \in \mathcal{N}_\infty^{k+1}$.*

*Proof.* As stated at the beginning of Appendix B, we can assume without loss of generality that $\boldsymbol{R}^\star = \boldsymbol{I}$ and $\boldsymbol{X}_i^\star = \boldsymbol{I}$ for all $1 \leq i \leq n$. We divide the index set $[n]$ into three sets

$$\mathcal{I}_1 = \left\{ i \mid \|\boldsymbol{X}_i^k - \boldsymbol{I}\|_F \leq \frac{\delta_k}{4} \right\}, \quad \mathcal{I}_2 = \left\{ i \mid \frac{\delta_k}{4} < \|\boldsymbol{X}_i^k - \boldsymbol{I}\|_F \leq \frac{3\delta_k}{4} \right\},$$

$$\text{and} \quad \mathcal{I}_3 = \left\{ i \mid \frac{3\delta_k}{4} < \|\boldsymbol{X}_i^k - \boldsymbol{I}\|_F \leq \delta_k \right\}.$$

For any $i \in \mathcal{I}_1 \bigcup \mathcal{I}_2$, we have

$$\|\boldsymbol{X}_i^k - \mu_k \widetilde{\nabla}_{\mathcal{R}} f(\boldsymbol{X}_i^k) - \boldsymbol{I}\|_F \leq \|\boldsymbol{X}_i^k - \boldsymbol{I}\| + \mu_k \|\widetilde{\nabla}_{\mathcal{R}} f(\boldsymbol{X}_i^k)\| \leq \frac{3\delta_k}{4} + 2\mu_k npq \leq \delta_{k+1},$$

$$\tag{49}$$

where the last inequality holds because we choose $\mu_k = \mathcal{O}(\frac{\delta_k}{n}) \leq \frac{\delta_k}{16n}$.

It remains to consider the case $i \in \mathcal{I}_3$. Firstly, it is easy to see that

$$\text{dist}(\boldsymbol{X}^k, \boldsymbol{X}^\star)^2 \geq \sum_{i \in \mathcal{I}_2 \bigcup \mathcal{I}_3} \|\boldsymbol{X}^k - \boldsymbol{X}^\star(\delta_k)\|_F^2 \geq \frac{\delta_k^2}{16}|\mathcal{I}_2 \bigcup \mathcal{I}_3|.$$

Note that we have $|\mathcal{I}_2 \bigcup \mathcal{I}_3| \leq \frac{\text{dist}(\boldsymbol{X}^k, \boldsymbol{X}^\star)^2}{\delta_k^2/16} \leq \frac{16\xi_k^2}{\delta_k^2} = \mathcal{O}(npq)$ according to the assumption $\xi_0 = \mathcal{O}(\sqrt{npq}\delta_0)$. Hence, for any $i \in \mathcal{I}_3$, we have

$$\widetilde{\nabla} f(\boldsymbol{X}_i) = \sum_{j \in \mathcal{A}_i \cap \mathcal{I}_1} \frac{\boldsymbol{X}_i - \boldsymbol{X}_j}{\|\boldsymbol{X}_i - \boldsymbol{X}_j\|_F} + \sum_{j \in \mathcal{A}_i \backslash \mathcal{I}_1} \frac{\boldsymbol{X}_i - \boldsymbol{X}_j}{\|\boldsymbol{X}_i - \boldsymbol{X}_j\|_F} + \widetilde{\nabla} h(\boldsymbol{X}_i). \tag{50}$$

Since $|\mathcal{I}_2 \bigcup \mathcal{I}_3| = \mathcal{O}(npq)$, we can choose $\xi_0$ properly such that $|\mathcal{I}_2 \bigcup \mathcal{I}_3| \leq \epsilon npq$. Then, we have

$$|\mathcal{A}_i \backslash \mathcal{I}_1| \leq |\mathcal{I}_2 \bigcup \mathcal{I}_3| \leq \epsilon npq \quad \text{and} \quad |\mathcal{A}_i \cap \mathcal{I}_1| \geq |\mathcal{A}_i| - |\mathcal{A}_i \backslash \mathcal{I}_1| \geq (1 - 2\epsilon)npq.$$

Let us use $\widetilde{\nabla} g^1(\boldsymbol{X}_i)$ to denote the first term on the RHS of (50). We have

$$\widetilde{\nabla} g^1(\boldsymbol{X}_i) = \sum_{j \in \mathcal{A}_i \cap \mathcal{I}_1} \frac{\boldsymbol{X}_i - \boldsymbol{X}_j}{\|\boldsymbol{X}_i - \boldsymbol{X}_j\|_F} = \sum_{j \in \mathcal{A}_i \cap \mathcal{I}_1} \frac{\boldsymbol{X}_i - \boldsymbol{I}}{\|\boldsymbol{X}_i - \boldsymbol{X}_j\|_F} + \frac{\boldsymbol{I} - \boldsymbol{X}_j}{\|\boldsymbol{X}_i - \boldsymbol{X}_j\|_F}$$

$$= \sigma(\boldsymbol{X}_i - \boldsymbol{I}) + \sum_{j \in \mathcal{A} \cap \mathcal{I}_1} \frac{\boldsymbol{I} - \boldsymbol{X}_j}{\|\boldsymbol{X}_i - \boldsymbol{X}_j\|_F}, \tag{51}$$

where $\sigma = \sum_{j \in \mathcal{A}_i \cap \mathcal{I}_1} \frac{1}{\|\boldsymbol{X}_i - \boldsymbol{X}_j\|_F}$. For the last term in the above equation, we have

$$\left\| \sum_{j \in \mathcal{A}_i \cap \mathcal{I}_1} \frac{\boldsymbol{I} - \boldsymbol{X}_j}{\|\boldsymbol{X}_i - \boldsymbol{X}_j\|_F} \right\|_F \leq \sum_{j \in \mathcal{A}_i \cap \mathcal{I}_1} \frac{\|\boldsymbol{I} - \boldsymbol{X}_j\|_F}{\|\boldsymbol{X}_i - \boldsymbol{X}_j\|_F} \leq \frac{\delta_k}{4} \sum_{j \in \mathcal{A}_i \cap \mathcal{I}_1} \frac{1}{\|\boldsymbol{X}_i - \boldsymbol{X}_j\|_F} = \frac{\delta_k \sigma}{4}.$$

In addition, the projection of $\boldsymbol{X}_i - \boldsymbol{I}$ onto the cotangent space can be bounded as

$$\|\mathcal{P}_{\text{T}_{\boldsymbol{X}_i}^\perp}(\boldsymbol{X}_i - \boldsymbol{I})\|_F = \|(\boldsymbol{X}_i - \boldsymbol{I})\|_F^2/2 \leq \delta_k^2/2,$$

which implies that $\widetilde{\nabla}_\mathcal{R} g^1(\boldsymbol{X}_i) = \mathcal{P}_{\text{T}_{\boldsymbol{X}_i}}(\widetilde{\nabla} g^1(\boldsymbol{X}_i))$ satisfies

$$\|\widetilde{\nabla}_\mathcal{R} g^1(\boldsymbol{X}_i) - \sigma(\boldsymbol{X}_i - \boldsymbol{I})\|_F \leq \frac{\delta_k \sigma}{4} + \frac{\delta_k^2 \sigma}{2} \leq \frac{\delta_k \sigma}{2}.$$

Then, the fact $\widetilde{\nabla}_\mathcal{R} f(\boldsymbol{X}_i) = \widetilde{\nabla}_\mathcal{R} g^1(\boldsymbol{X}_i) + \mathcal{P}_{\text{T}_{\boldsymbol{X}_i}^\perp}\left(\sum_{j \in \mathcal{A}_i \backslash \mathcal{I}_1} \frac{\boldsymbol{X}_i - \boldsymbol{X}_j}{\|\boldsymbol{X}_i - \boldsymbol{X}_j\|_F}\right) + \widetilde{\nabla}_\mathcal{R} h(\boldsymbol{X}_i)$ implies

$$\|\widetilde{\nabla}_\mathcal{R} f(\boldsymbol{X}_i) - \sigma(\boldsymbol{X}_i - \boldsymbol{I})\|_F \leq \|\widetilde{\nabla}_\mathcal{R} g^1(\boldsymbol{X}_i) - \sigma(\boldsymbol{X}_i - \boldsymbol{I})\|_F + |\mathcal{A}_i \backslash \mathcal{I}_1| + \left\|\widetilde{\nabla}_\mathcal{R} h(\boldsymbol{X}_i)\right\|_F$$

$$\leq \frac{\delta_k \sigma}{2} + \epsilon npq + 5\sqrt{2\delta_0}nq.$$

Next, motivated by the update of ReSync, we can construct

$$\boldsymbol{X}_i^k - \mu_k \widetilde{\nabla}_\mathcal{R} f(\boldsymbol{X}_i^k) - \boldsymbol{I} = (1 - \mu_k \sigma)(\boldsymbol{X}_i^k - \boldsymbol{I}) + \mu_k(\widetilde{\nabla}_\mathcal{R} f(\boldsymbol{X}_i) - \sigma(\boldsymbol{X}_i - \boldsymbol{I})),$$

which implies

$$\|\boldsymbol{X}_i^k - \mu_k \widetilde{\nabla}_\mathcal{R} f(\boldsymbol{X}_i^k) - \boldsymbol{I}\|_F \leq (1 - \mu_k \sigma)\|\boldsymbol{X}_i^k - \boldsymbol{I}\| + \mu_k \left(\frac{\delta_k \sigma}{4} + \frac{\delta_k^2 \sigma}{2} + \epsilon npq + 5\sqrt{2\delta_0}nq\right)$$

$$\leq (1 - \mu_k \sigma)\delta_k + \mu_k \left(\frac{\delta_k \sigma}{2} + \epsilon npq + 5\sqrt{2\delta_0}nq\right)$$

$$= \delta_k - \mu_k \left(\frac{\delta_k \sigma}{2} - \epsilon npq - 5\sqrt{2\delta_0}nq\right). \tag{52}$$

In order to further upper bound the above inequality, we can compute

$$\sigma = \sum_{j \in \mathcal{A}_i \cap \mathcal{I}_1} \frac{1}{\|\boldsymbol{X}_i - \boldsymbol{X}_j\|_F} \geq \sum_{j \in \mathcal{A}_i \cap \mathcal{I}_1} \frac{1}{\|\boldsymbol{X}_i - \boldsymbol{I}\|_F + \|\boldsymbol{X}_j - \boldsymbol{I}\|_F}$$

$$\geq \frac{4}{5\delta_k}|\mathcal{A}_i \cap \mathcal{I}_1| = \frac{4(1 - 2\epsilon)npq}{5\delta_k}.$$

where the second inequality holds because $\|\boldsymbol{X}_i - \boldsymbol{I}\|_F + \|\boldsymbol{X}_j - \boldsymbol{I}\|_F \leq 5\delta_k/4$ as $j \in \mathcal{I}_1$. It implies

$$\frac{\delta_k \sigma}{2} - \epsilon npq - 5\sqrt{2\delta_0}nq \geq \frac{2(1-2\epsilon)npq}{5} - \epsilon npq - 5\sqrt{2\delta_0}nq \geq \frac{npq}{4}.$$

By plugging the above bound into (52) and ignoring the high-order error term in (45), we complete the proof. $\qquad\square$

Finally, we present Lemma 10 and its proof.

**Lemma 10.** *With high probability, the following holds for all $\boldsymbol{X}^0 \in \mathcal{N}_\infty^0$:*

$$\left\| \widetilde{\nabla}_{\mathcal{R}} h(\boldsymbol{X}_i) \right\|_F \leq 5\sqrt{2\delta_0}nq \quad \forall i \in [n]. \tag{53}$$

*Proof of Lemma 10.* Firstly, define $\tilde{\boldsymbol{O}}_{ij} = \boldsymbol{O}_{ij}\boldsymbol{X}_j\boldsymbol{X}_i^\top$, then

$$\widetilde{\nabla} h(\boldsymbol{X}_i) = \sum_{j \in \mathcal{E}_i/\mathcal{A}_i} \frac{\boldsymbol{X}_i - \boldsymbol{O}_{ij}\boldsymbol{X}_j}{\|\boldsymbol{X}_i - \boldsymbol{O}_{ij}\boldsymbol{X}_j\|_F} = \sum_{j \in \mathcal{E}_i/\mathcal{A}_i} \frac{\boldsymbol{I} - \boldsymbol{O}_{ij}\boldsymbol{X}_j\boldsymbol{X}_i^\top}{\|\boldsymbol{X}_i - \boldsymbol{O}_{ij}\boldsymbol{X}_j\|_F}\boldsymbol{X}_i = \sum_{j \in \mathcal{E}_i/\mathcal{A}_i} \frac{\boldsymbol{I} - \tilde{\boldsymbol{O}}_{ij}}{\|\boldsymbol{I} - \tilde{\boldsymbol{O}}_{ij}\|_F}\boldsymbol{X}_i$$

The fact that $\boldsymbol{X} \in \mathcal{N}_\infty^0$ implies that $\|\boldsymbol{X}_i - \boldsymbol{X}_j\|_F \leq 2\delta_0$, i.e., $\|\boldsymbol{I} - \boldsymbol{X}_j\boldsymbol{X}_i^\top\|_F \leq 2\delta_0$. For any $\|\boldsymbol{O}_{ij} - \boldsymbol{I}\| \geq \sqrt{2\delta_0}$, we have

$$\left\| \frac{\boldsymbol{I} - \tilde{\boldsymbol{O}}_{ij}}{\|\boldsymbol{I} - \tilde{\boldsymbol{O}}_{ij}\|_F} - \frac{\boldsymbol{I} - \boldsymbol{O}_{ij}}{\|\boldsymbol{I} - \boldsymbol{O}_{ij}\|_F} \right\|_F \leq \left\| \frac{\boldsymbol{I} - \tilde{\boldsymbol{O}}_{ij}}{\|\boldsymbol{I} - \tilde{\boldsymbol{O}}_{ij}\|_F} - \frac{\boldsymbol{I} - \tilde{\boldsymbol{O}}_{ij}}{\|\boldsymbol{I} - \boldsymbol{O}_{ij}\|_F} \right\|_F$$

$$+ \left\| \frac{\boldsymbol{I} - \tilde{\boldsymbol{O}}_{ij}}{\|\boldsymbol{I} - \boldsymbol{O}_{ij}\|_F} - \frac{\boldsymbol{I} - \boldsymbol{O}_{ij}}{\|\boldsymbol{I} - \boldsymbol{O}_{ij}\|_F} \right\|_F \leq 2\sqrt{2\delta_0}$$

where the last inequality holds because

$$\left\| \frac{\boldsymbol{I} - \tilde{\boldsymbol{O}}_{ij}}{\|\boldsymbol{I} - \tilde{\boldsymbol{O}}_{ij}\|_F} - \frac{\boldsymbol{I} - \tilde{\boldsymbol{O}}_{ij}}{\|\boldsymbol{I} - \boldsymbol{O}_{ij}\|_F} \right\|_F = \|\boldsymbol{I} - \tilde{\boldsymbol{O}}_{ij}\|_F \left| \frac{1}{\|\boldsymbol{I} - \tilde{\boldsymbol{O}}_{ij}\|_F} - \frac{1}{\|\boldsymbol{I} - \boldsymbol{O}_{ij}\|_F} \right|$$

$$= \left| 1 - \frac{\|\boldsymbol{I} - \tilde{\boldsymbol{O}}_{ij}\|_F}{\|\boldsymbol{I} - \boldsymbol{O}_{ij}\|_F} \right|$$

$$= \frac{\left| \|\boldsymbol{I} - \boldsymbol{O}_{ij}\|_F - \|\boldsymbol{I} - \tilde{\boldsymbol{O}}_{ij}\|_F \right|}{\|\boldsymbol{I} - \boldsymbol{O}_{ij}\|_F} \leq \frac{\|\boldsymbol{O}_{ij} - \tilde{\boldsymbol{O}}_{ij}\|_F}{\|\boldsymbol{I} - \boldsymbol{O}_{ij}\|_F}$$

$$= \frac{\|\boldsymbol{I} - \boldsymbol{X}_j\boldsymbol{X}_i^\top\|_F}{\|\boldsymbol{I} - \boldsymbol{O}_{ij}\|_F} \leq \sqrt{2\delta_0},$$

and

$$\left\| \frac{\boldsymbol{I} - \tilde{\boldsymbol{O}}_{ij}}{\|\boldsymbol{I} - \boldsymbol{O}_{ij}\|_F} - \frac{\boldsymbol{I} - \boldsymbol{O}_{ij}}{\|\boldsymbol{I} - \boldsymbol{O}_{ij}\|_F} \right\|_F = \frac{\|\boldsymbol{O}_{ij} - \tilde{\boldsymbol{O}}_{ij}\|_F}{\|\boldsymbol{I} - \boldsymbol{O}_{ij}\|_F} = \frac{\|\boldsymbol{I} - \boldsymbol{X}_j\boldsymbol{X}_i^\top\|_F}{\|\boldsymbol{I} - \boldsymbol{O}_{ij}\|_F} \leq \sqrt{2\delta_0}.$$

Let $\Phi_i = \{j \in \mathcal{E}_i/\mathcal{A}_i \mid \|\boldsymbol{O}_{ij} - \boldsymbol{I}\| \leq \sqrt{2\delta_0}\}$. According to the fact that $|\mathcal{E}_i/\mathcal{A}_i| \leq (1+\epsilon)nq$ and the randomness of $\boldsymbol{O}_{ij}$, it is easy to show that $|\Phi_i| \leq \sqrt{2\delta_0}nq$ hold for all $1 \leq i \leq n$ with high probability. Thus, by splitting the sum $\sum_{j \in \mathcal{E}_i/\mathcal{A}_i}$ in to two parts: $j \in \Phi_i$ and $j \notin \Phi_i$, we have

$$\left\| \sum_{j \in \mathcal{E}_i/\mathcal{A}_i} \frac{\boldsymbol{I} - \tilde{\boldsymbol{O}}_{ij}}{\|\boldsymbol{I} - \tilde{\boldsymbol{O}}_{ij}\|_F} - \sum_{j \in \mathcal{E}_i/\Omega_i} \frac{\boldsymbol{I} - \boldsymbol{O}_{ij}}{\|\boldsymbol{I} - \boldsymbol{O}_{ij}\|_F} \right\|_F \leq 4\sqrt{2\delta_0}nq. \tag{54}$$

Besides, since $\boldsymbol{O}_{ij}$ is uniformly distributed on $\mathrm{SO}(d)$, according to Lemma A.1 in [39], we know that $\mathbb{E}\left\{ \frac{\boldsymbol{I} - \boldsymbol{O}_{ij}}{\|\boldsymbol{I} - \boldsymbol{O}_{ij}\|_F} \right\} = c(d)\boldsymbol{I}$. Then, the matrix Bernstein's inequality [38] tells us that, with high probability,

$$\left\| \sum_{j \in \mathcal{E}_i/\Omega_i} \frac{\boldsymbol{I} - \boldsymbol{O}_{ij}}{\|\boldsymbol{I} - \boldsymbol{O}_{ij}\|_F} - |\mathcal{E}_i/\Omega_i| \cdot c(d)\boldsymbol{I} \right\|_F \leq \sqrt{2\delta_0}nq. \tag{55}$$

This, together with (54), implies that

$$\left\| \sum_{j \in \mathcal{E}_i/\mathcal{A}_i} \frac{\boldsymbol{I} - \tilde{\boldsymbol{O}}_{ij}}{\|\boldsymbol{I} - \tilde{\boldsymbol{O}}_{ij}\|_F} - |\mathcal{E}_i/\Omega_i| \cdot c(d)\boldsymbol{I} \right\|_F \leq 5\sqrt{2\delta_0}nq.$$

The fact that $\widetilde{\nabla} h(\boldsymbol{X}_i) = \sum_{j \in \mathcal{E}_i/\mathcal{A}_i} \frac{\boldsymbol{I} - \tilde{\boldsymbol{O}}_{ij}}{\|\boldsymbol{I} - \tilde{\boldsymbol{O}}_{ij}\|_F} \boldsymbol{X}_i$ implies that

$$\left\| \widetilde{\nabla} h(\boldsymbol{X}_i) - |\mathcal{E}_i/\Omega_i| \cdot c(d)\boldsymbol{X}_i \right\|_F \leq 5\sqrt{2\delta_0}nq. \tag{56}$$

We complete the proof by taking the projection operator $\widetilde{\nabla}_{\mathcal{R}} h(\boldsymbol{X}_i) = \mathcal{P}_{\mathrm{T}\boldsymbol{X}_i}(\widetilde{\nabla} h(\boldsymbol{X}_i))$ and the fact that $\mathcal{P}_{\mathrm{T}\boldsymbol{X}_i}(\boldsymbol{X}_i) = 0$. $\qquad\square$

