# OpenReview forum: "ReSync: Riemannian Subgradient-based Robust Rotation Synchronization"
_NeurIPS.cc/2023/Conference — NeurIPS 2023 poster_

### Official Review · Reviewer_aU9r · 2023-07-01

**Soundness:** 3 good
**Presentation:** 3 good
**Contribution:** 3 good
**Rating:** 7
**Confidence:** 3

**Summary:**

The paper concerns synchronization of observed rotations with incomplete and corrupted observations. The authors construct the method ReSync that is a gradient-based algorithm for solving the problem. The paper describes the context, prior results on the synchronization problem, and the algorithm, and presents a thorough convergence analysis together with experimental evaluation.

**Strengths:**

- well-written and clearly presented paper
- the presented method deals with an important problem
- the presentation of the algorithm is followed by a thorough convergence analysis
- the method performs well in the experimental validation

**Weaknesses:**

- constructing a gradient descent algorithm is not a big contribution in itself. However, I believe the geometric setting and the connection to the theoretical analysis, which is not trivial, makes the contribution important

**Questions:**

no questions

**Limitations:**

yes

---

> ### Author Rebuttal · Authors · 2023-08-09
>
> We would like to thank the reviewer for the supportive and valuable comments. Should the reviewer have any further concerns, please inform us during the reviewer-author discussion period so that we can respond timely.

---

> > ### Comment · Reviewer_aU9r · 2023-08-12
> >
> > Thanks for the reply. My scoring has not changed.

---

> > > ### Author Response · Authors · 2023-08-18
> > > **Thank you for your response**
> > >
> > > Thank you very much for reading our rebuttal and for your response. We will be closely following the Reviewer-Author discussion period in case the reviewer has any additional concerns or questions.

---

### Official Review · Reviewer_grLa · 2023-07-03

**Soundness:** 3 good
**Presentation:** 3 good
**Contribution:** 3 good
**Rating:** 7
**Confidence:** 3

**Summary:**

This paper presents a theoretical study for robust rotation synchronization with a least-unsquared minimization formulation over the rotation group.

In particular, this paper proposes a two-step algorithm called ReSync, where the first step uses spectral initialization to generate an initial guess and the second step performs Riemannian Subgradient descent from the initial guess.

The paper proves that, under suitable conditions of the random corruption model, this algorithm converges linearly to the groundtruth rotations.

The paper presents numerical experiments that verify the correctness of the theorem and compares the performance of ReSync with other state-of-the-art algorithms.

**Strengths:**

- The theoretical contribution of this paper advances the previous state of the art.
- Paper is well written and easy to follow, despite being a theory paper.

**Weaknesses:**

- I am curious if similar guarantees could be made in the case where the inlier measurements are corrupted by small (and bounded) noise? Could you guarantee the algorithm converges to a solution that has bounded error from the groundtruth rotations?

**Questions:**

See Weaknesses.

---

> ### Author Rebuttal · Authors · 2023-08-09
>
> We would like to thank the reviewer for the supportive and valuable comments. We address the concern below.
>
> **A. Guarantees with additive noise.** Yes. It is possible to show the convergence results against additive noise, in which the algorithm will converge to the neighborhood of the ground-truth rotations up to the scale of the noise. The initialization analysis should not change much as it fits well with additive noise. Then, a crucial step involves providing a noise-perturbed version of the weak sharpness property, i.e.,
> $$
> f(\boldsymbol{X}) - f^\star \geq \mathcal{O}(npq) \operatorname{dist}_1(\boldsymbol{X},\boldsymbol{X}^\star) - \nu,
> $$
>
> where the perturbation $\nu$ is caused by additive noise. The main difficulty lies in conducting the contraction analysis, as the presence of additive noise brings additional challenges for controlling the infinity norm-induced distance. We believe it is possible to circumvent this technical difficulty by studying more carefully the property of the noise. This line of research is our ongoing work.
>
> We hope that our response is satisfactory to the reviewer and that the concern has been addressed appropriately. Should the reviewer have any further concerns, please inform us during the reviewer-author discussion period so that we can respond timely.

---

> > ### Comment · Reviewer_grLa · 2023-08-18
> >
> > Thanks for the response, I maintain my original score.

---

> > > ### Author Response · Authors · 2023-08-18
> > > **Thank you for your response**
> > >
> > > Thank you very much for reading our rebuttal and for your response. We will be closely following the Reviewer-Author discussion period in case the reviewer has any additional concerns or questions.

---

### Official Review · Reviewer_W8A3 · 2023-07-09

**Soundness:** 3 good
**Presentation:** 3 good
**Contribution:** 3 good
**Rating:** 7
**Confidence:** 4

**Summary:**

This work proposes to solve the rotation synchronization problem using Riemannian subgradient method with spectral initialization. The proposed formuation is sum of absolute deviations, which is robust to outliers. Exact recovery guarantees are provided under uniform corruption model (the graph is Erdos Renyi and probability of corruption 1-p). Numerical results show competitive performance of the proposed method compared to other state-of-the-art methods.

**Strengths:**

1. The theoretical result in the noiseless case (corruption only) is quite strong. That is, it shows linearly convergence to the ground truth rotations whenever n>1/(p^7q^2) up to a log factor, where p is the probability of being a clean edge (conditioned on being an edge), and q is the probability of being an edge.
2. The numerical experiments show advantages over previous state-of-the-art methods in the presense of both corruption and noise.
3. The proofs look correct.

Overall, I enjoyed reading the paper.

**Weaknesses:**

1. This is not necessarily a weakness, but it would be even nicer if the authors could comment on the stability of your algorithm to noise (would it be possible to show an approximate recovery in this case)?



**Questions:**

I wonder how sensitive is your method to initialization? For example, given random initialization, what is the typical behavior of your algorithm in numerical experiments and what about the theoretical results?



**Limitations:**

Yes

---

> ### Author Rebuttal · Authors · 2023-08-09
>
> We would like to thank the reviewer for the supportive and valuable comments. We address the concerns in a point-by-point manner below.
>
> **A. Stability to noise.** Yes. It is possible to show stability and convergence results against additive noise, which is our ongoing work. The initialization analysis should not change much as it fits well with additive noise. Then, a crucial step involves providing a noise-perturbed version of the weak sharpness property, i.e.,
> $$
>     f(\boldsymbol{X}) - f^\star \geq \mathcal{O}(npq) \operatorname{dist}_1(\boldsymbol{X},\boldsymbol{X}^\star) - \nu,
> $$
>
> where the perturbation $\nu$ is caused by additive noise. The main difficulty lies in conducting the contraction analysis, as the presence of additive noise brings additional challenges for controlling the infinity norm-induced distance. We believe it is possible to circumvent this technical difficulty by studying more carefully the property of the noise. The final result will be convergence to the neighborhood of the ground-truth rotations up to the scale of the noise.
>
> **B. Sensitivity to initialization.** We test our algorithm with random initialization using the setting of Fig. 2(a) in our manuscript; see Fig. 2 in the one-page supplementary PDF of the rebuttal. It can be observed that our algorithm continues to work. However, it requires a much larger diminishing factor $\gamma$ of the step size, resulting in notably slower convergence. Theoretically, we do not have an idea yet of how to prove convergence guarantees with random initialization, given that the geometric property (e.g., weak sharpness) only holds locally.
>
> We hope that our response is satisfactory to the reviewer and that all concerns have been addressed appropriately. Should the reviewer have any further concerns, please inform us during the reviewer-author discussion period so that we can respond timely.

---

> > ### Comment · Reviewer_W8A3 · 2023-08-22
> >
> > I thanks authors for the response and it addressed all my questions. After reading all the reviews and comments, it does not change my opinion that this is a solid theoretical paper. Therefore, I prefer not to change my score.

---

> > > ### Author Response · Authors · 2023-08-22
> > > **Thank you for your response**
> > >
> > > Thank you so much for reading all the reviews and our rebuttals and for your response.

---

### Official Review · Reviewer_m1vD · 2023-07-29

**Soundness:** 1 poor
**Presentation:** 2 fair
**Contribution:** 2 fair
**Rating:** 6
**Confidence:** 4

**Summary:**

The paper proposes a Riemannian subgradient based algorithm for the robust rotation synchronization (RRS) problem. RRS involves recovering the absolute rotations of objects from the possibly corrupted/noisy relative rotations between pairs of objects. The problem setting involves two ratios: q \in [0,1] denotes the observation ratio and p \in [0,1] denotes the true observation ratio. The paper pose the problem as a (non-convex and non-smooth) least-unsquared minimization formulation over the rotation group. The proposed method ReSync has a spectral relaxation based initialization procedure, which is followed by Riemannian subgradient iterations. The main contribution of the paper is to show that under random corruption model (RCM) setting: (a) the proposed initialization (X^0) can be relatively close to true solution (X^*), depending on p and q, and (b) given the initialization guarantee, the Riemannian subgradient descent show local linear rate of convergence. Overall, the paper show that ReSync converges linearly to the ground-truth rotations when p^7 q^2 = \Omega(log n / n). Towards the end of the draft, the paper has few experimental results that compare the proposed algorithm against state-of-the-art.

**Strengths:**

The paper presents an interesting approach for recovering ground truth (X^*) in RRS problem. The key theoretical guarantees for ReSync comes from a) an initialization procedure SpectrIn, which ensures that the initialization X^0 is close to X^*, b) weak sharpness property of the least unsquared formulation, which is being solved via ReSync, and c) local linear convergence analysis for ReSync based on initialization and weak sharpness property.

I have, however, not verified the correctness of the theoretical results.

**Weaknesses:**

Concerns regarding theory:
1. The paper assumes missing observations as zero matrix, which does not lie on the SO(d) manifold. Hence, only if (i,j) belongs to available observation, Y_{ij} \in SO(d). The paper does not provide any justification for this choice. A more suitable choice seems to be Identity matrix as it lies on SO(d).
2. In line 178-180, it is stated that E[Y_ij] = pq X_i^*(X_j^*)^\top for all (i,j). This does not seem correct as it is not clear how outlier points O_ij \in SO(d) are handled while computing this expectation.
3. While the paper cites and discusses its differences with [27] in lines 171-175, it seems that [27] should be discussed in more detail. While  [27] focuses on orthogonal group with additive Gaussian noise and permutation group with outliers, it should be noted that permutation group is a special subset of orthogonal group. Interestingly, [27] states that "though it is not analyzed in our manuscript, the proof technique for the permutation group synchronization under uniform corruption could be directly modified to tackle this O(d) synchronization under uniform multiplicative corruption" (in the paragraph before Section 3.2). The proof of leave-one-out technique seems to be adopted from [27]. Hence, while [27] has been cited in Section 3.1 of the paper, it does not seem to be the main contribution and could been discussed in the supplementary material. Overall, [27] deserves more discussion, especially w.r.t. the above quoted statement, and in this regard contribution of the paper should be clearly highlighted.
4. A discussion on computational cost of the proposed algorithm is missing.

Concerns regarding experiments:
1. The paper shows only a few empirical results on synthetic datasets. While this gives some insights on how the algorithm works in lab environment, performance on real-world setting gives an idea on how the algorithm will perform when used in real applications. If space was a factor, the paper could have moved some of the proofs/proof-outlines to the supplementary section.
2. While the paper discusses [40] and states that it "introduces a least-unsquared formulation and applies the SDR method to tackle it", the paper should have mentioned more directly that the main formulation (2), which papers tries to solve, was originally proposed in [40]. Hence, while the theoretical results of [40] are in q=1 setting, the paper should have empirically compared with [40] as well. Similarly, paper [30] should also be compared empirically.
3. Experiments are done in two settings: with and without additive noise. Without additive noise setting has been theoretically analyzed in the paper. One question is that in this setting, DESC method seems to be better or similar to ReSync. Any insights as to why it does better (where it does)?

**Questions:**

Please look in the weakness section.

---

> ### Author Rebuttal · Authors · 2023-08-09
>
> We would like to thank the reviewer for the valuable comments. We address the concerns in a point-by-point manner below.
>
> **A. Missing observations (Q1 in concerns regarding theory).** Since formulation (2) only relies on available observations, we are allowed to assign the missing entries as $\mathbf{0}$. In theory, this setting ensures that the $d$ leading eigenvectors of $\mathbb{E}(\boldsymbol{Y})$ are $\boldsymbol{X}^\star$. However, assigning missing observations as identity matrices may not necessarily achieve this outcome when $q$ is small. This particular choice has also been utilized in previous works; see the overview in the fourth paragraph of [24, Section 2.1] (here, [24] refers to reference [24] in our manuscript). In experiments, we use the missing entries only in spectral initialization (Algorithm 2). We have also conducted a simulation to illustrate that assigning the missing observations as identity matrices will decrease the performance; see Fig. 1 in the one-page supplementary PDF of the rebuttal.
>
> We shall add the following sentence in line 29 in the revised version: *``The missing observations are set to be $\mathbf{0}$ by convention; see, e.g., [24, Section 2.1].''*
>
> **B. Computation of Expectation (Q2 in concerns regarding theory).** We are sorry for the confusion caused. We have used the fact $\mathbb{E} (\boldsymbol{O}_{ij}) = \mathbf{0}$ since outliers are assumed to be independently and uniformly distributed on $\operatorname{SO}(d)$ in the RCM. In the revised version, we shall replace "our random graph setup" in line 178 with *"the RCM"* to enhance clarity.
>
> **C. Connections to [27] and highlight contributions (Q3 in concerns regarding theory).** We have two major differences to [27]: 1) Nontrivial modifications due to the specific structure of $\operatorname{SO}(d)$. Our approach follows the standard leave-one-out analysis based on the standard "Dist" (up to $\operatorname{O}(d)$ invariance) defined above Lemma 3 in the Appendix. Nonetheless, we have to transfer the results to "dist" due to the structure of $\operatorname{SO}(d)$ in Lemma 5, which is new. 2) Handling incomplete observation ($q<1$). In the case of incomplete observation, the constructed $\boldsymbol{W}$ in (17) in the Appendix becomes more intricate; it has the additional third column, rendering the analysis of our Lemma 2 more involved. We shall elaborate on these discussions in the revised version.
>
> Concerning our contributions, in addition to the above nontrivial modifications in the initialization analysis, our more important contributions are found in Sections 3.2 and 3.3. These sections focus on geometric and contraction analyses, respectively, which were not present in [27].
>
> **D. Computational cost (Q4 in concerns regarding theory).** Algorithm 2 has computational cost $\mathcal{O}(n^3)$.  The per-iteration complexity of the Riemannian subgradient procedure is $\mathcal{O}(n^2 q)$.  We shall add these discussions in Section 2.1 in the revised version.
>
> **E. Lack of real-world experiment results (Q1 in concerns regarding experiments).** Since the primary focus of this work is theory, we originally only conducted synthetic experiments to corroborate our theoretical findings. Per the request of the reviewer, we now implemented the experiment in [40, Fig. 7] for the real-world "Lucy dataset" (here, [40] refers to reference [40] in our manuscript); see Fig. 5 in the one-page supplementary PDF of the rebuttal. It can be observed that our method outperforms DESC and LUD (here, LUD refers to the algorithm in [40] and we use their implementation and default parameters). We shall add this experiment result in our revised version.
>
> Moreover, we are studying applying our algorithm to Cryo-EM imaging experiment settings (based on common-lines), which is our ongoing work.
>
> **F. References [40] and [30] (Q2 in concerns regarding experiments).** We shall add the following sentence at the beginning of line 63 in the revised version: *"This formulation was introduced in [40] as the initial step for applying the SDR method."*
>
> We have compared our algorithm with LUD in [40] (we use their implementation and default parameters) in the setting of Fig. 2 in our manuscript; see Fig. 3 in the one-page supplementary PDF of the rebuttal. LUD has competitive performance when additive noise is present, which is reasonable since LUD attempts to solve a convex relation of problem (2). We shall add these comparisons in the revised version. We do not compare with [30], as we observed that it has slightly suboptimal performance compared to CEMP and MPLS (which we included in our experiments) in its own simulation results; see [30, Fig. 4].
>
> **G. Discussion on DESC (Q3 in concerns regarding experiments).**  In the absence of additive noise, the good-cycle condition in DESC is likely fulfilled. This, together with its carefully designed post-procedure for recovering the ground-truth rotations from the estimated corruption level, leads to highly competitive experiment performance.
>
> In fact, our method performs better if we use larger $n$; see Fig. 4 in the one-page supplementary PDF of the rebuttal, where we use the setting of Fig. 3(a) and 3(c) in our manuscript with the only difference being $n =500$ rather than $n = 200$.
>
> We hope that our response is satisfactory to the reviewer and that all concerns have been addressed appropriately. Should the reviewer have any further concerns, please inform us during the reviewer-author discussion period so that we can respond timely.

---

> > ### Comment · Reviewer_m1vD · 2023-08-22
> > **Response to rebuttal**
> >
> > I thank the authors for the detailed response.
> >
> > Regarding point A: The following point is not clear. It would be nice if the authors could elaborate on this.
> > >In theory, this setting ensures that the  leading eigenvectors of $\mathbb{E}[\mathbf{Y}]$ are $\mathbf{X^{*}}$.
> >
> > Regarding point B: It would be nice if the authors could explain why independently and uniformly distributed outliers on SO(d) will have zero mean.
> > >We have used the fact $\mathbb{E}[\mathbf{O}_{ij}]=0$ since outliers are assumed to be independently and uniformly distributed on SO(d) in the RCM.
> >
> > The authors have answered my other questions. I have accordingly changed my score

---

### Author Rebuttal · Authors · 2023-08-09

Dear ACs and Reviewers,

This global response contains our one-page supplementary PDF of the rebuttal. All additional figures are included in this file. Please find it in the attachment.

Best regards,

Authors.

---

### Decision · Program_Chairs · 2023-09-21

**Decision:**

Accept (poster)

**Comment:**

The paper deals with the robust rotation synchronization problem. It presents an initialization strategy and algorithm with a theoretical analysis. Overall, the reviewers and the AC think that the paper merits acceptance. Please consider the reviews and make the necessary modifications.